# A miR-327–FGF10–FGFR2-mediated autocrine signaling mechanism controls white fat browning

Carina Fischer[1], Takahiro Seki[1], Sharon Lim[1], Masaki Nakamura[1], Patrik Andersson[1], Yunlong Yang[1], Jennifer Honek[1], Yangang Wang[2], Yanyan Gao[2], Fang Chen[3], Nilesh J. Samani[4], Jun Zhang[5], Masato Miyake[5], Seiichi Oyadomari[5], Akihiro Yasue[6], Xuri Li[7], Yun Zhang[8], Yizhi Liu[7] & Yihai Cao[1,2,8]

Understanding the molecular mechanisms regulating beige adipocyte formation may lead to the development of new therapies to combat obesity. Here, we report a miRNA-based autocrine regulatory pathway that controls differentiation of preadipocytes into beige adipocytes. We identify miR-327 as one of the most downregulated miRNAs targeting growth factors in the stromal-vascular fraction (SVF) under conditions that promote white adipose tissue (WAT) browning in mice. Gain- and loss-of-function experiments reveal that miR-327 targets FGF10 to prevent beige adipocyte differentiation. Pharmacological and physiological β-adrenergic stimulation upregulates FGF10 levels and promotes preadipocyte differentiation into beige adipocytes. In vivo local delivery of miR-327 to WATs significantly compromises the beige phenotype and thermogenesis. Contrarily, systemic inhibition of miR-327 in mice induces browning and increases whole-body metabolic rate under thermoneutral conditions. Our data provide mechanistic insight into an autocrine regulatory signaling loop that regulates beige adipocyte formation and suggests that the miR-327–FGF10–FGFR2 signaling axis may be a therapeutic targets for treatment of obesity and metabolic diseases.

[1] Department of Microbiology, Tumor and Cell Biology, Karolinska Institute, 171 77 Stockholm, Sweden. [2] Department of Endocrinology, Affiliated Hospital of Qingdao University, Qingdao 266003, China. [3] Hospital of Zhejiang Chinese Medicine University, 54 Youdian Road, Hangzhou, Zhejiang 310006, China. [4] Department of Cardiovascular Sciences, University of Leicester and NIHR Leicester Cardiovascular Biomedical Research Unit, Glenfield Hospital, Leicester LE3 9QP, UK. [5] Division of Molecular Biology, Institute for Genome Research, Institute of Advanced Medical Sciences, Tokushima University, 3-18-15 Kuramoto-cho, Tokushima 770-8503, Japan. [6] Department of Orthodontics Dentofacial Orthopedics, Institute of Biomedical Sciences, Tokushima University Graduate School, 3-18-15 Kuramoto-cho, Tokushima 770-8504, Japan. [7] State Key Laboratory of Ophthalmology, Zhongshan Ophthalmic Center, Sun Yat-Sen University, Guangzhou 510060, China. [8] The Key Laboratory of Cardiovascular Remodeling and Function Research, Chinese Ministry of Education and Chinese Ministry of Public Health, Shandong University Qilu Hospital, Jinan, Shandong 250012, China. Correspondence and requests for materials should be addressed to Y.C. (email: yihai.cao@ki.se)

White adipose tissue (WAT) browning-triggered thermogenesis is an emerging mechanism for energy consumption and provides a novel therapeutic option for the treatment of obesity and metabolic disorders, such as type 2 diabetes mellitus (T2DM)[1–7]. Cold exposure, β3-adrenoceptor agonists, diet, drugs, and many other stimuli are able to activate brown adipose tissue (BAT) and to induce WAT browning[8]. Depending on their anatomical location, different WAT depots possess intrinsic properties that define their ability to remain white or become beige adipocytes[9, 10]. In response to mild cold exposure, rodent subcutaneous WAT (scWAT) manifests a "brown-like" phenotype, whereas visceral WAT (visWAT), a genuine WAT, is resistant to browning at similar temperatures[9, 11–14]. WAT browning-triggered thermogenesis may play an important role to increase whole-body energy expenditure and results in reduced fat mass, increased insulin sensitivity and improved blood lipid profiles[15]. However, the importance of WAT browning physiology remains a controversial issue[8].

In the adipose tissue microenvironment (AME), non-adipocytes including vascular cells, inflammatory cells, and mesenchymal stromal cells play crucial roles in modulation of adipocyte metabolism[16]. For example, endothelial cells, perivascular cells in the vessel wall and mesenchymal stromal cells have potentials to differentiate into adipocytes[2, 3, 17]. We recently demonstrated that endothelial cells produce paracrine factors to modulate adipocyte functions and thereby play a determinant role in controlling preadipocyte differentiation and adipocyte browning through a paracrine regulatory mechanism[12]. Therefore, complex interactions between different cell types in the AME collectively control adipocyte function and metabolic activity. Soluble factors that mediate cellular interactions between various cell types remain a challenging issue although adipocyte-derived factors and cytokines are relatively well characterized[3]. Furthermore, molecular regulation of growth factor production under various physiological and pathological conditions is still unknown.

MicroRNAs (miRNAs) are short non-coding RNA molecules that regulate protein expression through post-transcriptional mechanisms[18]. They often act as repressors of protein production by controlling the mRNA levels and inhibiting translation of their targets within a cell. Most miRNA studies in adipose tissues are aimed to study miRNA expression profiles in lean and obese WAT of animals and humans, and some miRNAs show strong correlations with obesity, insulin resistance and T2DM. Additionally, many studies show that numerous genes are differentially expressed in scWAT and visWAT[19]. However, miRNAs and their targets regulating WAT browning are largely unknown. In particular, targeting miRNAs in non-adipocytes during WAT browning is a novel approach.

Fibroblast growth factors encompass probably the largest growth factor family among all known growth factor families[20]. In the human FGF family, there are 22 structurally and functionally related members that display a broad range of biological functions including embryonic development, wound healing, angiogenesis, stem cell differentiation, and endocrine regulation[21]. FGFs bind to four FGF receptors (FGFRs) to initiate signaling events that mediate these biological functions in target cells[22]. FGFR1 and FGFR2 appear to be the most commonly distributed growth factor receptors and they mediate most biological functions of the FGF family. FGF10 is a secreted protein and genetic deletion of the Fgf10 gene in mice results in impaired lung development and absence of limb bud formation[23–25]. Furthermore, the development of abdominal scWAT is markedly impaired in FGF10-deficient mice[26]. Tissue distribution studies showed that the adipose tissue expresses the most abundant Fgf10 transcript among all adult tissues. Moreover, FGF10 expression is particularly enriched in the stromal-vascular fraction (SVF) of WAT where adipocyte progenitors reside[27].

In this work, we identify miR-327 as a key regulator that controls preadipocyte differentiation toward beige adipocytes. We show that cold exposure and β3-adrenoceptor activation markedly downregulate miR-327 expression in the browning WAT-SVF. Moreover, FGF10, as a direct target of miR-327, is increased in these settings and controls preadipocyte to beige adipocyte differentiation. In vitro and in vivo gain- and loss-of-function experiments validate the key function of miR-327 in adipocyte differentiation and WAT browning. Our data identify a regulatory pathway in non-adipocytes of the SVF consisting of miR-327, FGF10, and FGFR2 that controls beige adipocyte differentiation through an autocrine mechanism. Modulation of this signaling pathway may provide a therapeutic option for the treatment of obesity and T2DM.

## Results

**Cold exposure and β3-adrenoceptor agonist induced WAT browning.** We aimed to define signaling molecules in SVF cells that control adipocyte differentiation. To achieve this goal, we chose a mouse WAT browning model for the following reasons: (1) During browning of WAT, there is a significant expansion of the vascular component; (2) differentiation of preadipocyte to beige adipocyte is one of the key processes of adipocyte browning[28]; (3) thermogenesis can be accurately measured; and (4) the effect of therapeutic interventions can be easily assessed. We took two approaches to induce a beige phenotype in WATs. First, a β3-adrenoceptor agonist CL-316243 (CL) was systemically administrated to adult C57BL/6 mice. After CL-treatment for 5 days, scWAT and visWAT exhibited distinct phenotypes of smaller adipocytes with dense intracellular contents (Fig. 1a, c). Immunohistochemical analysis showed that CL-treated mitochondria had enriched COX4[+] staining in scWAT and visWAT, indicating an increased number of mitochondria in these white adipose depots (Fig. 1a, d). Moreover, CL-treated WATs showed increased levels of UCP1, the key thermogenic protein, with a more pronounced expression in scWAT compared to visWAT (Fig. 1a, e).

To define the SVF components, endothelial cell-specific marker CD31 and the perivascular stromal fibroblast marker alpha-smooth muscle actin (αSMA) were used for immunohistochemical analysis. Along with the WAT browning, markedly augmented angiogenesis concomitantly occurred in CL-treated scWAT and visWAT (Fig. 1a, f). Under CL treatment, both scWAT and visWAT showed increased αSMA signals especially around the arterioles (Fig. 1a, g). Like CL treatment, exposure of mice to cold temperature exacerbated a browning phenotype of scWAT, demonstrating smaller adipocytes, high mitochondrial COX4 content, high level of UCP1, angiogenesis, and robust expression of αSMA (Fig. 1b, h–l). However, cold-induced increases of mitochondria and UCP1 were significantly less pronounced in visWAT (Fig. 1b, i, j). These findings are consistent with the general view that scWAT undergoes a beige transition under cold exposure whereas visWAT is rather resistant to cold-induced browning.

**miR-327 is downregulated under β3-adrenonegic stimulation and targets Fgf10.** To define miRNAs and their targeted signaling molecules in the SVF during early browning, we performed a genome-wide miRNA profiling analysis. CL treatment induced alterations in expression levels of a range of miRNAs in the WAT-SVF (Fig. 2a). We focused on downregulated miRNAs because, we were interested to identify upregulated targeted signaling molecules that might mediate an SVF-adipocyte crosstalk.

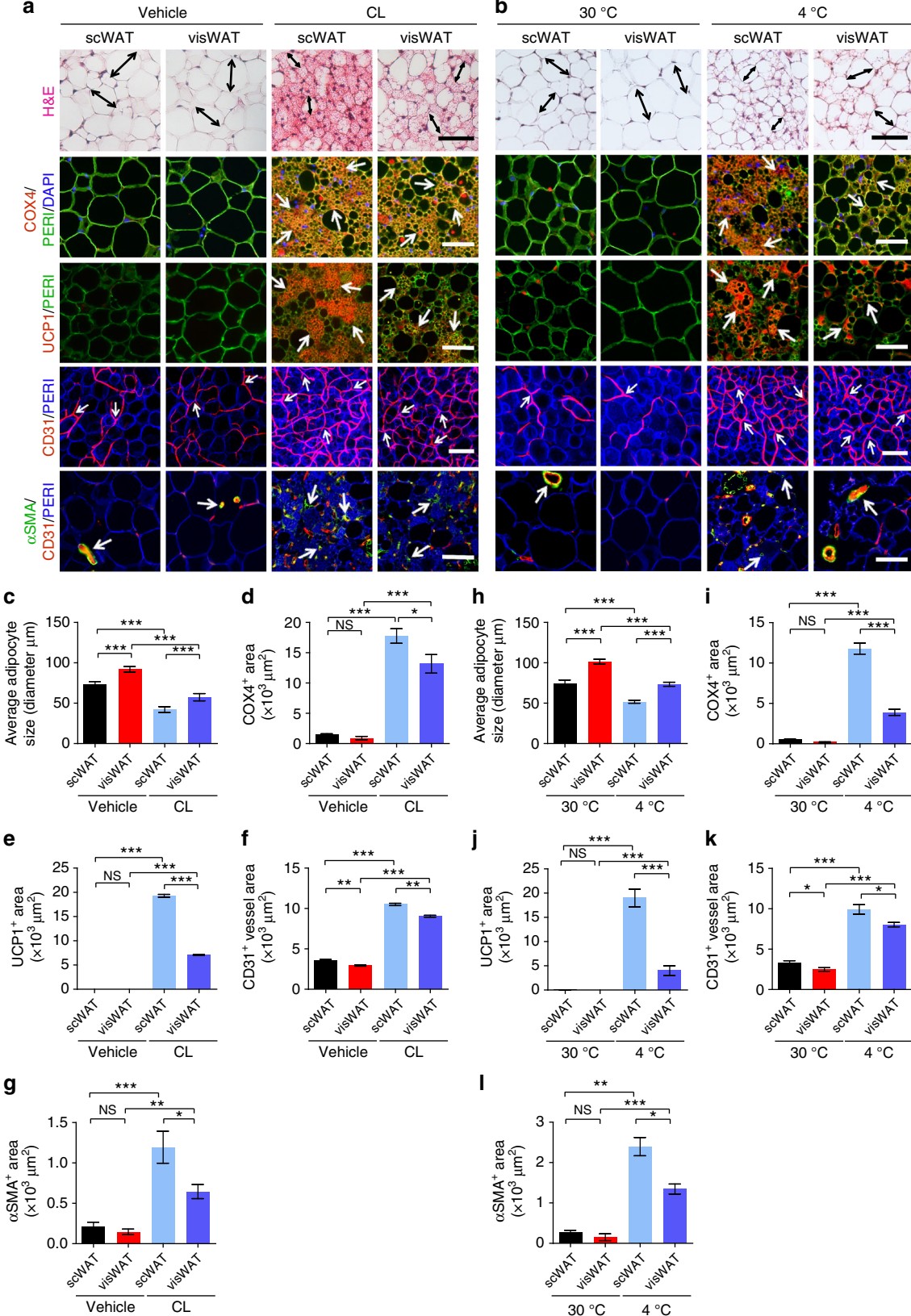

**Fig. 1** CL and low temperature lead to WAT browning, angiogenesis, and an increased number of myofibroblast-like cells. **a**, **b** Histological analysis of adipocyte morphology (H&E), adipocytes (PERI), mitochondria (COX4), uncoupling protein 1 (UCP1), blood vessels (CD31), and myofibroblast-like cells (αSMA) in **a** 5-day CL-316243-treated scWAT and visWAT compared to vehicle-treated control. **b** Two-week 4 °C-treated scWAT and visWAT compared to 30 °C control. Double-headed arrows mark adipocyte diameter. Arrows point to respective positive signals. **c–l** Quantifications of adipocyte size and positive signals per field of COX4, UCP1, CD31, and αSMA of CL-316243- and vehicle-, and 30 °C- and 4 °C- treated scWATs and visWATs (>30 adipocytes per field; $n = 10$ random fields; $n = 5$ mice per group). PERI, perilipin; COX4, mitochondrial complex 4; UCP1, uncoupling protein 1. Scale bars, 100 μm. NS, not significant. *$P < 0.05$, **$P < 0.01$, and ***$P < 0.001$ by Student's $t$-test. Data presented as mean ± s.e.m.

Among significantly downregulated miRNAs, miR-327 was a novel and stably downregulated miRNA (Fig. 2a, b, Supplementary Fig. 1). miR-327 was markedly downregulated in CL-treated SVFs isolated from scWAT and visWAT as quantitatively analyzed by qPCR analysis (Fig. 2c). A nearly 10-fold decrease of miR-327 was observed in CL-treated SVFs isolated from scWAT

and visWAT relative to their vehicle-treated controls. At day 7 after CL treatment, miR-327 remained unchanged at a very low level similar to day 1 of treatment (Supplementary Fig. 1). By contrast, no decrease of miR-327 was found in the SVFs derived from iBAT isolated from CL-treated mice (Fig. 2d), suggesting that the pattern of miR-327 downregulation was restricted to

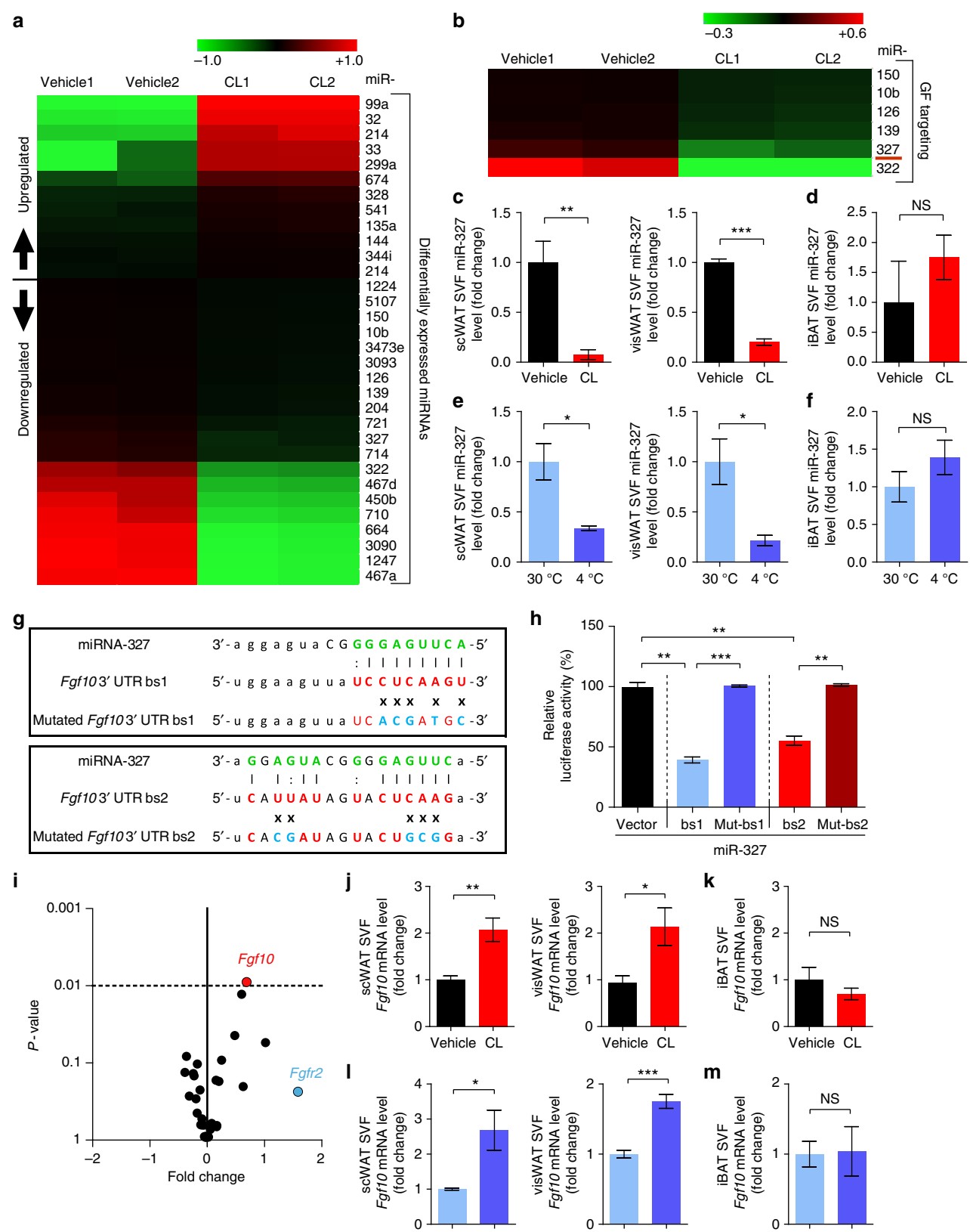

SVFs derived from WAT. Like CL treatment, cold exposure also markedly decreased miR-327 expression levels in SVFs isolated from scWAT and visWAT, but not from iBAT (Fig. 2e, f).

Target analysis showed that miR-327 targets the 3′-untranslated region (UTR) of *Fgf10* mRNA with two binding sites (Fig. 2g). To validate *Fgf10* mRNA as a target, we cloned 300-base-pair (bp) fragments upstream and downstream of the *Fgf10* bs1 or *Fgf10* bs2 into a psiCHECK-2 vector containing a cytomegalovirus (CMV)-promoter-luciferase reporter. To validate miR-327-specific binding, five nucleotide substitutions were introduced into each binding site, generating mutated *Fgf10* bs1 and mutated *Fgf10* bs2, which were non-complementary to miR-327. Co-transfection of human embryonic kidney 293 (HEK293T) cells with psiCHECK-2-*Fgf10* bs1, or psiCHECK-2-*Fgf10* bs2 and miR-327 mimic resulted in significant inhibition of luciferase activity (Fig. 2h). Conversely, co-transfection of a miR-327 mimic with mutated psiCHECK-2-*Fgf10* bs1 or mutated psiCHECK-2-*Fgf10* bs2 completely attenuated the inhibition of luciferase activity (Fig. 2h).

**Upregulation of FGF10 in browning WAT-SVF.** Genome-wide mRNA profiling of the WAT-SVF showed that FGF10 was the most significantly upregulated member within the FGF family (Fig. 2i). Additionally, FGFR2 was markedly upregulated in the SVF of browning WAT. Consistent with these findings, qPCR analysis further validated that increased levels of *Fgf10* mRNA were present in the SVFs of browning scWAT and visWAT, but not in iBAT (Fig. 2j, k). Almost identical results were obtained in cold-induced browning WAT, showing significant upregulation of FGF10 in browning WATs, but not in iBAT (Fig. 2l, m). These data demonstrate that FGF10 upregulation is linked to WAT browning. Furthermore, expression of *Fgf10* mRNA was rather restricted to the SVF (Supplementary Fig. 2).

To provide further experimental evidence, we measured miR-327, *Fgf10*, and *Fgfr2* in scWAT, visWAT and iBAT derived mature adipocytes under thermoneutral and cold conditions. Expression levels of miR-327, *Fgf10*, and *Fgfr2* remained unchanged in adipocytes derived from cold-exposed animals compared to thermoneutral controls (Supplementary Fig. 3a–d).

We next analyzed FGF10 protein levels in WATs and iBAT under various experimental conditions. FGF10 protein was considerably increased in browning scWAT and visWAT after only 3 days of CL-treatment. In visWAT, FGF10 was practically undetectable in the vehicle-treated control but accumulated upon CL treatment. In scWAT, a more than sevenfold increase of FGF10 protein was detected in CL-treated tissue relative to its control (Fig. 3a, d, Supplementary Fig. 10a). In contrast to visWAT, the basal level of FGF10 protein in scWAT control samples was detectable in most samples, but at a rather low level. We further analyzed cold-exposed scWAT and visWAT and could show a marked increase in FGF10 protein level compared to their respective 30 °C-exposed controls (Fig. 3b, e,

Supplementary Fig. 10b). It appeared that increases of FGF10 protein levels in browning scWAT and visWAT were greater than the increases in mRNA levels (Figs 2, 3). In agreement with the mRNA expression levels, CL- and cold-activated iBAT did not show significant alterations of FGF10 protein levels (Fig. 3c, f, Supplementary Fig. 10c). These findings further validated that FGF10 was significantly increased in CL- and cold-induced browning scWAT and visWAT, but not in iBAT.

**Cellular localization of FGF10, miR-327, and FGFR2 and release of FGF10 protein.** To gain detailed information about the cellular source of FGF10, we isolated different cell types from the SVF of scWAT. These cell types included CD45$^+$ immune cells (T cells, monocytes, leukocytes, dendritic cell, and NK cells), CD31$^+$ vascular endothelial cells, PDGFR-α$^+$ stromal fibroblasts, PDGFR-α$^-$ cells and mature adipocytes. Interestingly, *Fgf10* mRNA levels were restricted to the PDGFR-α$^+$ cellular faction. CD45$^+$, CD31$^+$, PDGFR-α$^-$ fractions and adipocyte fractions expressed significantly lower levels of *Fgf10* mRNA (Fig. 3g). Furthermore, expression levels of *Fgfr2* mRNA and miR-327 in the PDGFR-α$^+$ fraction were 4–20-fold higher than those in other cell fractions (Fig. 3h, i). These findings suggest the existence of an autocrine regulatory loop in PDGFR-α$^+$ stromal fibroblasts.

To study the release of FGF10 protein, we isolated primary preadipocytes from the subcutaneous adipose depot. Conditioned medium or cell lysate were collected from primary cell cultures and FGF10 protein levels measured. As shown in Fig. 3j, an ~sevenfold higher FGF10 protein level was detected in the conditioned medium relative to its intracellular amount. These findings demonstrate that FGF10 is effectively exported from its producing cells.

**Target validation of the miR-327–FGF10 axis.** To further validate FGF10 as a target of miR-327, 3T3-L1 preadipocytes were used for in vitro studies. 3T3-L1 preadipocytes expressed considerably high basal levels of *Fgf10* mRNA and FGF10 protein (Fig. 4a, b). To identify possible functional impacts of miR-327 levels on preadipocyte differentiation, a miR-327 mimic and a miR-327 inhibitor were used to transfect 3T3-L1 preadipocytes. These molecular tools effectively altered miR-327 levels in these cells (Supplementary Fig. 3e). Importantly, transfection of 3T3-L1 preadipocytes with a miR-327 mimic, but not with a miRNA control, significantly downregulated *Fgf10* mRNA and FGF10 protein levels (Fig. 4a–c, Supplementary Fig. 11a), validating the finding that FGF10 is a direct target of miR-327. Conversely, transfection of 3T3-L1 preadipocytes with a miR-327-inhibitor, but not with a miRNA control, significantly increased *Fgf10* mRNA and FGF10 protein levels (Fig. 4d–f, Supplementary Fig. 11b). To experimentally investigate the FGF signaling, 3T3-L1 preadipocytes were transfected with the miR-327 mimic or a miRNA control (miR-NC). Phosphorylation of FGFR substrate 2-alpha (FRS2-α), a biomarker for FGFR activation[29, 30], was significantly inhibited in

**Fig. 2** Identification of miR-327 and FGF10 as SVF-derived modulators of early browning. **a** Heatmap of significantly ($P < 0.05$) up- and downregulated miRNAs from 3-day CL-316243-treated visWAT-SVFs compared to vehicle-treated controls ($n = 5$ mice per sample; $n = 2$ samples per group). **b** Heatmap of significantly downregulated growth factor (GF)-targeting miRNAs extracted from **a**. **c–f** qPCR analysis of miR-327 in **c**, **d** 3-day CL-316243-treated SVFs or **e**, **f** 1-week 4 °C-treated SVFs of scWAT, visWAT, and iBAT compared to respective controls. Sno-202 served as an internal control ($n = 5$ samples per group). **g** Alignment details of miR-327 to *Fgf10* 3′ UTR binding site 1 (bs1) and binding site 2 (bs2). Xs mark nucleotides in bs1 and bs2 that were mutated to generate Mut-bs1 and Mut-bs2. **h** Quantification of *Renilla* luciferase activity normalized to Firefly luciferase activity in miR-327 mimic-treated HEK293T cells co-transfected with psiCHECK-2 vector, psiCHECK-2-*Fgf10* bs1, psiCHECK$^{TM}$-2-*Fgf10* bs2, psiCHECK$^{TM}$-2-*Fgf10* Mut-bs1, or psiCHECK$^{TM}$-2-*Fgf10* Mut-bs2 ($n = 5$ samples per group). **i** Volcano plot of the FGF ligand and receptor families from genome-wide microarray analysis of 3-day CL-316243-treated visWAT-SVF. Lowest $P$-value is highlighted in red and highest fold-change is highlighted in blue ($n = 4$ mice per sample; $n = 3$ samples per group). **j–m** qPCR analysis of *Fgf10* in **j**, **k** 3-day CL-316243-treated SVF or **l**, **m** 1-week 4 °C-treated SVF of scWAT, visWAT, and iBAT compared to respective controls ($n = 5$ samples per group). NS, not significant. *$P < 0.05$, **$P < 0.01$, and ***$P < 0.001$ by Student's *t*-test. Data presented as mean ± s.e.m.

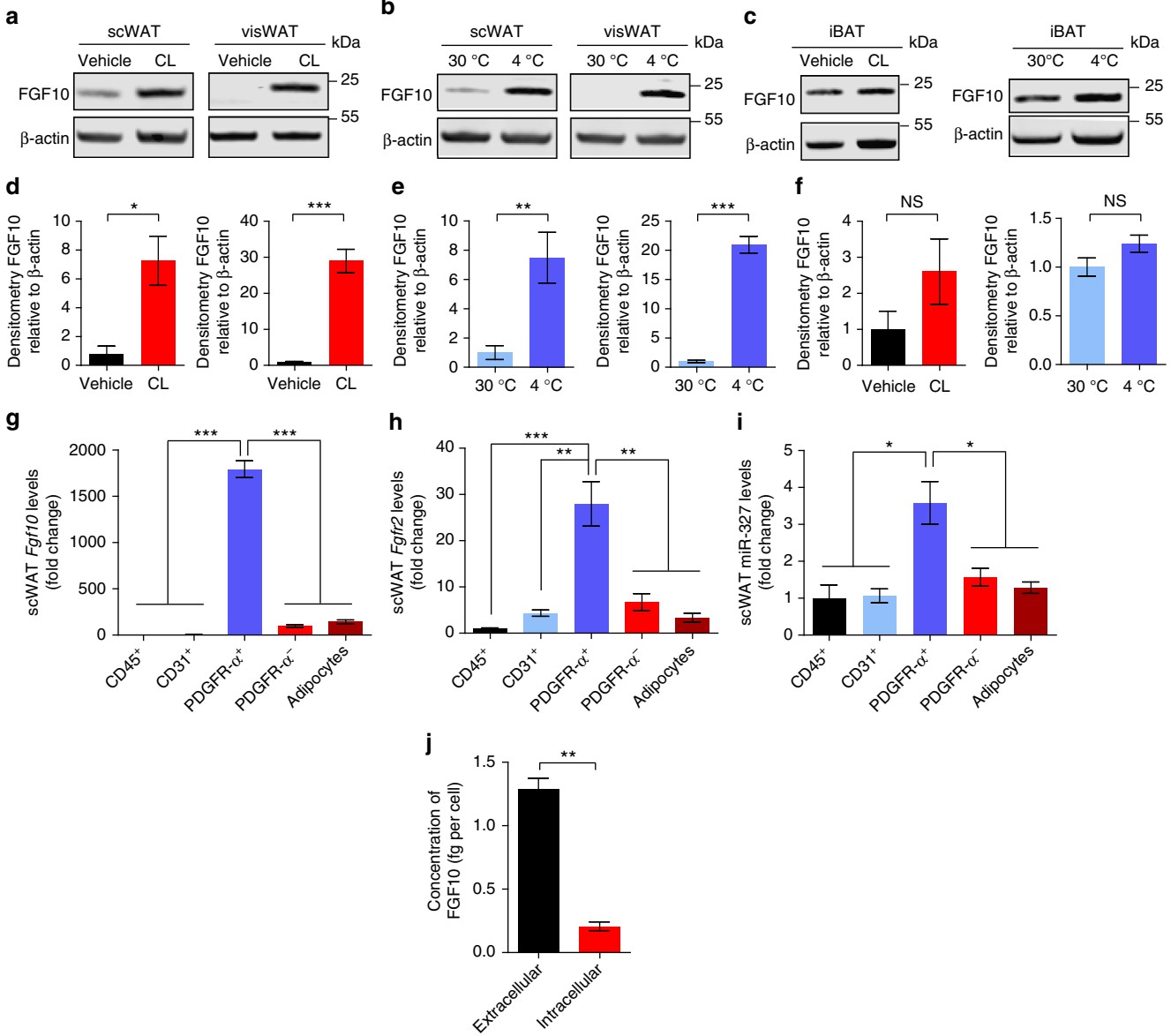

**Fig. 3** FGF10 protein is upregulated upon browning and secreted by preadipocytes. **a–f** Western immunoblot analysis of FGF10 in 3-day CL-316243- or 1-week 4 °C-treated scWAT, visWAT, and iBAT compared to their respective controls. FGF10 protein levels were quantified as densitometric signals and normalized to β-actin ($n = 5$ samples per group). **g–i** qPCR analysis of *Fgf10, Fgfr2*, and miR-327 mRNA levels in different cellular fractions of scWAT ($n = 5$ samples per group). **j** ELISA analysis of FGF10 protein levels in primary preadipocytes isolated from scWAT ($n = 5$ samples per group). kDa, kilodalton. NS, not significant. $*P < 0.05$, $**P < 0.01$, and $***P < 0.001$ by Student's $t$-test. Data presented as mean ± s.e.m.

the miR-327 mimic-transfected 3T3-L1 preadipocytes (Fig. 4g, h, Supplementary Fig. 11c). We further demonstrated that treatment with the miR-327 mimic markedly inhibited phosphorylation of Akt, whereas the total amount of Akt protein was not altered (Fig. 4i, j, Supplementary Fig. 11d). Notably, phosphorylation of Erk was not significantly altered in the miR-327 mimic- compared to the miR-NC-transfected 3T3-L1 preadipocytes (Fig. 4k, l, Supplementary Fig. 11e). These findings validated that FGF10 is a target of miR-327 that inhibits the FGF10-triggered FGFR–Akt, but not FGFR–Erk, signaling pathway (Fig. 4m).

**miR-327 modulates FGF10–FGFR2-dependent preadipocyte differentiation**. As FGFR–Akt signaling has been shown to be involved in adipocyte differentiation[31, 32], inhibition of FGFR–Akt activation by miR-327 suggested that miR-327 might target preadipocyte differentiation. To study the functional

impact of miR-327 on adipocyte differentiation, 3T3-L1 pre-adipocytes were induced for differentiation[33–35]. Surprisingly, transfection of 3T3-L1 preadipocytes with a miR-327 mimic, but not with miR negative control (miR-NC), largely blocked adi-pocyte differentiation as shown by deposition of Oil Red O⁺ lipid droplets (Fig. 5a). Quantification analysis showed an ~90% inhibition of preadipocyte differentiation in the miR-327 transfected 3T3-L1 preadipocytes relative to miR-NC trans-fected cells (Fig. 5c). In a gain-of-function experimental setting, we modified the differentiation protocol by lowering insulin levels. With the modified protocol only ~35% of preadipocytes differentiated to adipocytes (Fig. 5b, d). Under such conditions, transfection of 3T3-L1 preadipocytes with a miR-327 inhibitor significantly increased the number of differentiated adipocytes (Fig. 5b, d).

To relate our findings to physiological relevance, we further isolated primary preadipocytes using a standard protocol[35].

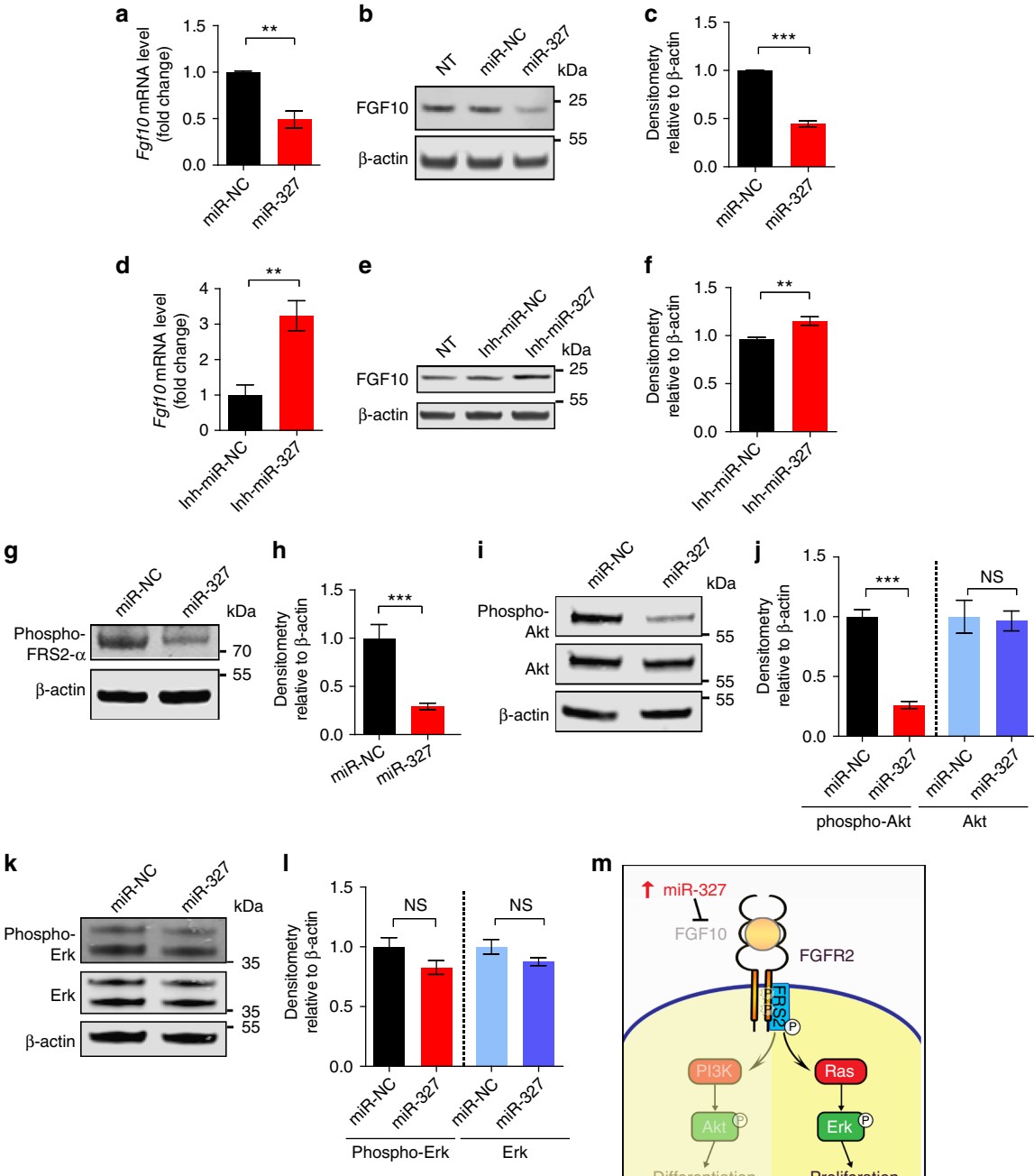

**Fig. 4** miR-327 mimics block FGF10–FGFR2-triggered PI3K–Akt signaling. **a** qPCR analysis of *Fgf10* in miR-negative control (miR-NC) and miR-327 mimic-treated 3T3-L1 preadipocytes ($n = 5$ samples per group). **b, c** Western immunoblot analysis and quantification of FGF10 in miR-NC and miR-327 mimic-treated 3T3-L1 preadipocytes ($n = 3$ samples per group). **d** qPCR analysis of *Fgf10* and **e, f** Western immunoblot analysis and quantification of FGF10 in inhibitor miR-negative control (Inh-miR-NC) and miR-327-inhibitor (Inh-miR-327)-treated 3T3-L1 preadipocytes ($n = 5$ samples per group). **g–l** Western immunoblot analysis and quantification of **g, h** phospho-FRS2-α, **i, j** phospho-Akt and Akt and **k, l** phospho-Erk and Erk in miR-NC and miR-327 mimic-treated 3T3-L1 cells ($n = 3$ samples per group). **m** Schematic diagram of the FGF10–FGFR2 signaling pathway for cell proliferation and differentiation. The Ras–Erk signaling is crucial for cell proliferation. The PI3K–Akt pathway is essential for cell differentiation. miR-327 specifically blocks the FGF10–FGFR2–triggered PI3K–Akt signaling during preadipocyte differentiation.). kDa, kilodalton. NS, not significant. *$P < 0.05$, **$P < 0.01$, and ***$P < 0.001$ by Student's *t*-test. Data presented as mean ± s.e.m.

Induction of scWAT-SVF cell to differentiation resulted in nearly 100% differentiation into mature adipocytes (Fig. 5e, f). Like the results obtained in 3T3-L1 preadipocytes, transfection of primary adipocytes with a miR-327 mimic largely blocked differentiation in these cells (Fig. 5e, f). We were further interested if transfection of primary preadipocytes with a miR-327 mimic blocks beige adipocyte differentiation. Upon stimulation with the non-selective β-adrenoceptor agonist isoproterenol, the miR-327

mimic-transfected cells showed reduced expression levels of differentiation markers and loss of browning and mitochondrial markers such as *Ucp1*, *Cidea*, *Ppargc1a*, and *Cox7a* (Fig. 5g, h, Supplementary Fig. 3f–i).

The fact that FGF10 is endogenously expressed in 3T3-L1 preadipocytes and that these preadipocytes efficiently differentiated into adipocytes suggested the existence of an autocrine regulatory loop in controlling adipocyte differentiation that

involves FGF10. Quantitative measurement of protein levels revealed that FGF10 was mainly localized in the extracellular fraction of preadipocytes and only modest levels were detected in mature adipocytes under various experimental settings (Supplementary Fig. 3j, k).

To study FGF10-dependent preadipocyte differentiation, we reduced the insulin level and exogenously added FGF10 recombinant protein in the differentiation medium. FGF10 significantly stimulated 3T3-L1-preadipocyte

differentiation (Fig. 6a, d). Quantification of Oil Red O+ differentiated adipocytes showed that a greater than twofold increase in differentiation was achieved in FGF10-stimulated 3T3-L1 preadipocytes (Fig. 6d). Transfection of miR-327 significantly inhibited 3T3-L1 preadipocyte differentiation under the reduced differentiation conditions and could be partly rescued by recombinant FGF10 treatment (Fig. 6a, d).

It is known that FGF10 binds to FGFR2b, which is expressed in preadipocytes[26]. Transfection of 3T3-L1 preadipocytes with

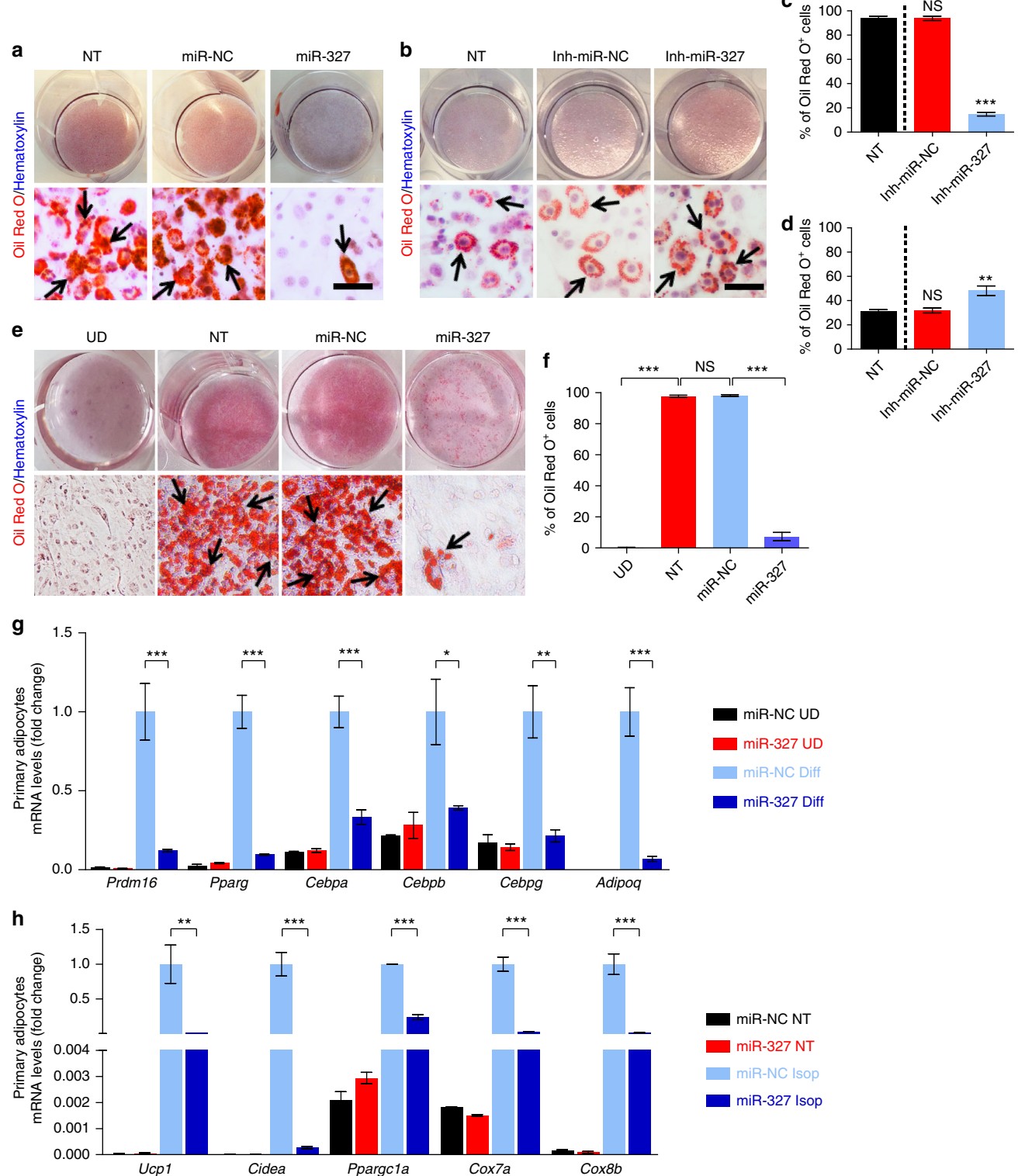

*Fgfr2*-siRNA (siFgfr2) almost completely blocked FGF10-induced differentiation (Fig. 6b, e). To functionally link FGF10-induced preadipocyte differentiation to the Akt pathway, we used an Akt inhibitor in our differentiation studies. Inhibition of Akt activity effectively inhibited FGF10-induced preadipocyte differentiation (Fig. 6c, f). These findings demonstrated that the FGFR2–Akt signaling pathway mediates FGF10-stimulated preadipocyte differentiation.

As FGF10 might affect preadipocyte proliferation, the impact of FGF10 on cell viability and proliferation were analyzed using preadipocytes treated with FGF10, siFgf10, miR-327, and miR-327 inhibitor before, during, or after differentiation. No significant difference of cell viability was seen before or during differentiation (Supplementary Fig. 4). Ki67$^+$ proliferating cell populations remained unchanged in all treated and non-treated samples (Supplementary Fig. 4). These findings demonstrate that FGF10 does not affect cell proliferation in our experimental settings.

We further validated our finding with a loss-of-function approach by knocking down *Fgf10* in preadipocytes. Introduction of a siRNA targeting *Fgf10* to 3T3-L1 preadipocytes effectively blocked differentiation (Fig. 6g, h), indicating that FGF10 is intrinsically required for preadipocyte differentiation. Importantly, Inh-miR-327 treatment is not sufficient to induce differentiation in siFgf10 co-tranfected cells (Fig. 6g, h). Addition of FGF10-rich conditioned medium derived from preadipocytes; however, rescued the inhibitory effect of siFgf10 (Fig. 6g, h). Similarly, transfection of preadipocytes with siFgfr2 also effectively blocked miR-327 inhibitor-induced cell differentiation (Fig. 6i, j). These data demonstrate that the FGF10–FGFR2 pathway is essential for preadipocyte differentiation and strengthen the finding that miR-327 specifically regulates FGF10 levels in preadipocytes and thereby controls adipocyte differentiation.

**In vivo delivery of a miR-327 mimic inhibits WAT browning.** To study the functional impact of miR-327 on WAT browning, we used an adenovirus expressing miR-327 (Ad-miR-327) for in vivo delivery. This vector also expressed GFP for monitoring of viral infection in tissues. Local delivery of Ad-miR-327 to scWATs and visWATs resulted in an effective infection of adipose tissues (Fig. 7a, Supplementary Fig. 5a). Consistent with the high infection efficacy, greater than 20-fold and ~8-fold increase in miR-327 levels were achieved by this in vivo delivery system in scWAT and visWAT, respectively, (Fig. 7b, Supplementary Fig. 5b).

We further examined the functional impact of miR-327 on WAT browning. First, CL- and cold-induced decreases in adipocyte size were significantly blocked after delivery of Ad-miR-327 in scWAT (Fig. 7c, d, g, h). Along with adipocyte size changes, COX4$^+$ mitochondrial contents were markedly decreased in Ad-miR-327 transfected scWAT relative to controls

(Fig. 7c, e, g, i). Importantly, UCP1 levels in CL- and cold-treated Ad-miR-327-infected scWATs were markedly decreased compared to their respective controls (Fig. 7c, f, g, j). Similar functional impairments were found in CL-treated Ad-miR-327-infected visWAT, albeit less pronounced browning phenotypes occurred in visWAT compared to those in scWAT (Supplementary Fig. 5). By contrast, cold exposure did not augment sufficient browning in visWAT and thus delivery of Ad-miR-327 did not produce any significant alterations in this adipose depot (Supplementary Fig. 5).

Like the in vitro findings, local delivery of Ad-miR-327 to scWAT ablated CL- and cold-induced FGF10 protein levels (Fig. 8a, c, d, f, Supplementary Fig. 12). Along with a decreased level of FGF10, Ad-miR-327 also effectively inhibited CL- and cold-induced UCP1 expression (Fig. 8b, c, e, g, Supplementary Fig. 12). Together, these data show that delivery of miR-327 to WATs markedly inhibits a browning phenotype in vivo through an FGF10-triggered pathway.

**Global metabolic changes in the miR-327 gain-of-function model.** To determine the overall phenotype of mice overexpressing miR-327 in scWAT and visWAT, body weight, food intake, tissue weight, glucose and insulin tolerance were measured (Fig 8h, i, Supplementary Fig. 6a–j). Detailed analysis of adipose depots showed that miR-327 mimic-treated scWAT fat mass was significantly reduced under both CL- and cold-treated conditions; however, treatment with Ad-miR-327 diminishes this effect significantly (Fig. 8h, i). Total body weight, food intake, glucose, and insulin tolerance were unchanged by miR-327 overexpression (Supplementary Fig. 6a–j).

In addition, metabolic rate, norepinephrine(NE)-induced oxygen consumption and core body temperature were measured to address if overexpression of miR-327 in scWAT and visWAT influence total energy expenditure. At 4 °C, miR-327 overexpression significantly reduced the metabolic rate and NE-induced oxygen consumption (Fig. 8k, l). In addition, Ad-miR-327 overexpression reduced the core body temperature in 4 °C exposed mice (Supplementary Fig. 6k). In contrast, metabolic rate, NE-induced oxygen consumption and core body temperature were similar between control and miR-327 overexpressing mice under thermoneutral condition (Fig. 8j, l, Supplementary Fig. 6k).

**In vivo delivery of a miR-327 inhibitor induces WAT browning.** To proof the concept of therapeutic potentials by targeting miR-327, we subcutaneously administered a miR-327 inhibitor to mice under different temperatures. Notably, delivery of the miR-327 inhibitor to mice housed at 30 °C induced a phenotypical change in scWAT, manifesting smaller adipocyte sizes and an increased expression level of the mitochondrial marker COX4 (Fig. 9a–c). UCP1 expression was detectable in smaller adipocytes, indicating browning and thermogenic activity

**Fig. 5** The functional impacts of miR-327 on preadipocyte differentiation. **a, b** Assessment of adipocyte differentiation by Oil Red O and hematoxylin double staining of non-treated (NT)-, miR-NC-, miR-327-, Inh-miR-NC-, and Inh-miR-327-treated 3T3-L1 cells. Arrows indicate differentiated Oil Red O$^+$ adipocytes. **c, d** Quantification of differentiated Oil Red O$^+$ adipocytes as percentages of total adipocytes (>30 adipocytes per field; $n = 10$ random fields). **e, f** Assessment and quantification of adipocyte differentiation by Oil Red O and hematoxylin double staining of undifferentiated (UD)-, non-treated (NT)-, miR-NC-, and miR-327-treated primary adipocytes derived from scWAT. Arrows indicate differentiated Oil Red O$^+$ adipocytes. Quantification of differentiated Oil Red O$^+$ adipocytes as percentages of total adipocytes (>30 adipocytes per field; $n = 10$ random fields). **g** qPCR analysis of mRNA levels of *Prdm16*, *Pparg*, *Cebpa*, *Cebpb*, *Cebpg*, and *Adipoq* in miR-NC undifferentiated (UD), miR-327 UD, miR-NC differentiated (Diff) and miR-327 Diff primary scWAT-SVF cells normalized to miR-NC Diff samples. *Actb* served as an internal control ($n = 5$ samples per group). **h** qPCR analysis of mRNA levels of *Ucp1*, *Cidea*, *Ppargc1a*, *Cox7a*, and *Cox8b* in miR-NC non-treated (NT), miR-327 NT, miR-NC isoproterenol (Isop)- and miR-327 Isop-treated primary adipocytes differentiated from scWAT-SVF cells. Samples were normalized using the miR-NC Isop group as a reference. *Actb* served as an internal control ($n = 5$ samples per group). Scale bars, 100 μm. NS, not significant. *$P < 0.05$, **$P < 0.01$, and ***$P < 0.001$ by Student's *t*-test. Data presented as mean ± s.e.m.

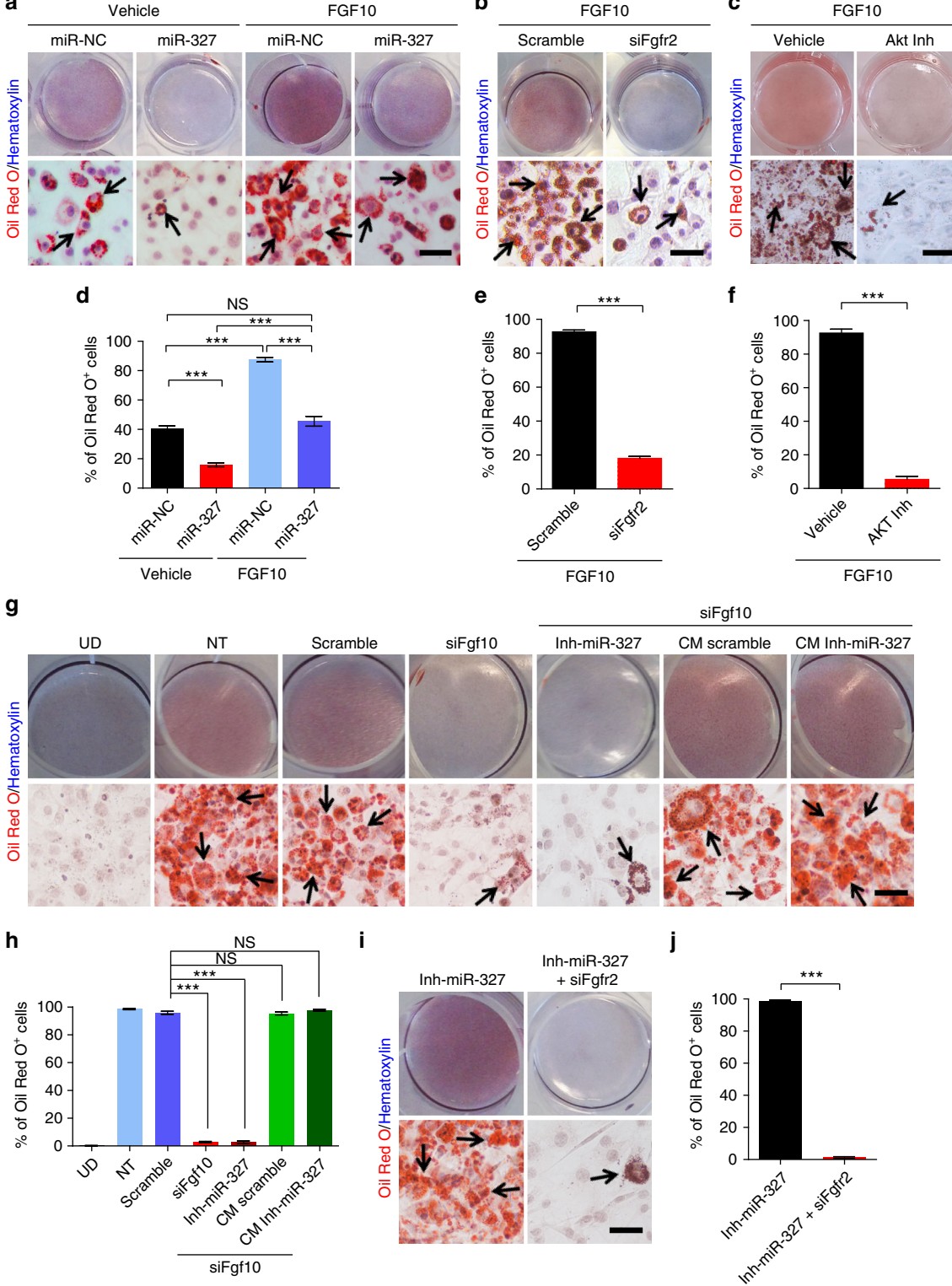

**Fig. 6** FGF10-induced FGFR2 activation is an essential process for preadipocyte differentiation. **a–c** The effects of miR-NC, miR-327, Scramble, siRNA-Fgfr2 (siFgfr2), and an Akt inhibitor (Akt Inh) on vehicle- and FGF10-induced 3T3-L1 preadipocyte differentiation visualized by Oil Red O and hematoxylin double staining. **d–f** Quantification of differentiated Oil Red O+ adipocytes as percentages of total adipocytes. **g** Differentiation of 3T3-L1 preadipocyte under various treated and non-treated conditions: Undifferentiated (UD); Non-treated (NT); Scramble; siFgf10-treated; siFgf10 plus Inh-miR-327; siFgf10 plus conditioned medium (CM) from Scramble-transfected preadipocytes; and siFgf10 plus conditioned medium (CM) from Inh-miR-327-transfected preadipocytes. **h** Quantification of average percentages of differentiated Oil Red O+ adipocytes vs. total cells. **i, j** Detection of adipocyte differentiation by Oil Red O+ and hematoxylin double staining of Inh-miR-327- and Inh-miR-327 plus siFgfr2-treated 3T3-L1 preadipocytes. **a–j** Arrows indicate differentiated Oil Red O+ adipocytes. Scale bars, 100 μm. Quantification of differentiated Oil Red O+ adipocytes as percentages of total adipocytes (>30 adipocytes per field; $n = 10$ random fields). NS, not significant. *$P < 0.05$, **$P < 0.01$, and ***$P < 0.001$ by Student's $t$-test. Data presented as mean ± s.e.m.

(Fig. 9a, d). These findings were unexpected because scWAT does not commonly undergo a browning transition under thermoneutral conditions. Under cold exposure, miR-327 inhibition markedly potentiated cold-induced browning (Fig. 9a–d). These findings show that delivery of a miR-327 inhibitor alone augments a browning phenotype of WAT and exacerbates the cold-induced beige transition.

Consistent with phenotypic changes, delivery of the miR-327 inhibitor to 30 °C-exposed mice robustly induced FGF10 protein levels in scWAT and delivery of the miR-327 inhibitor during

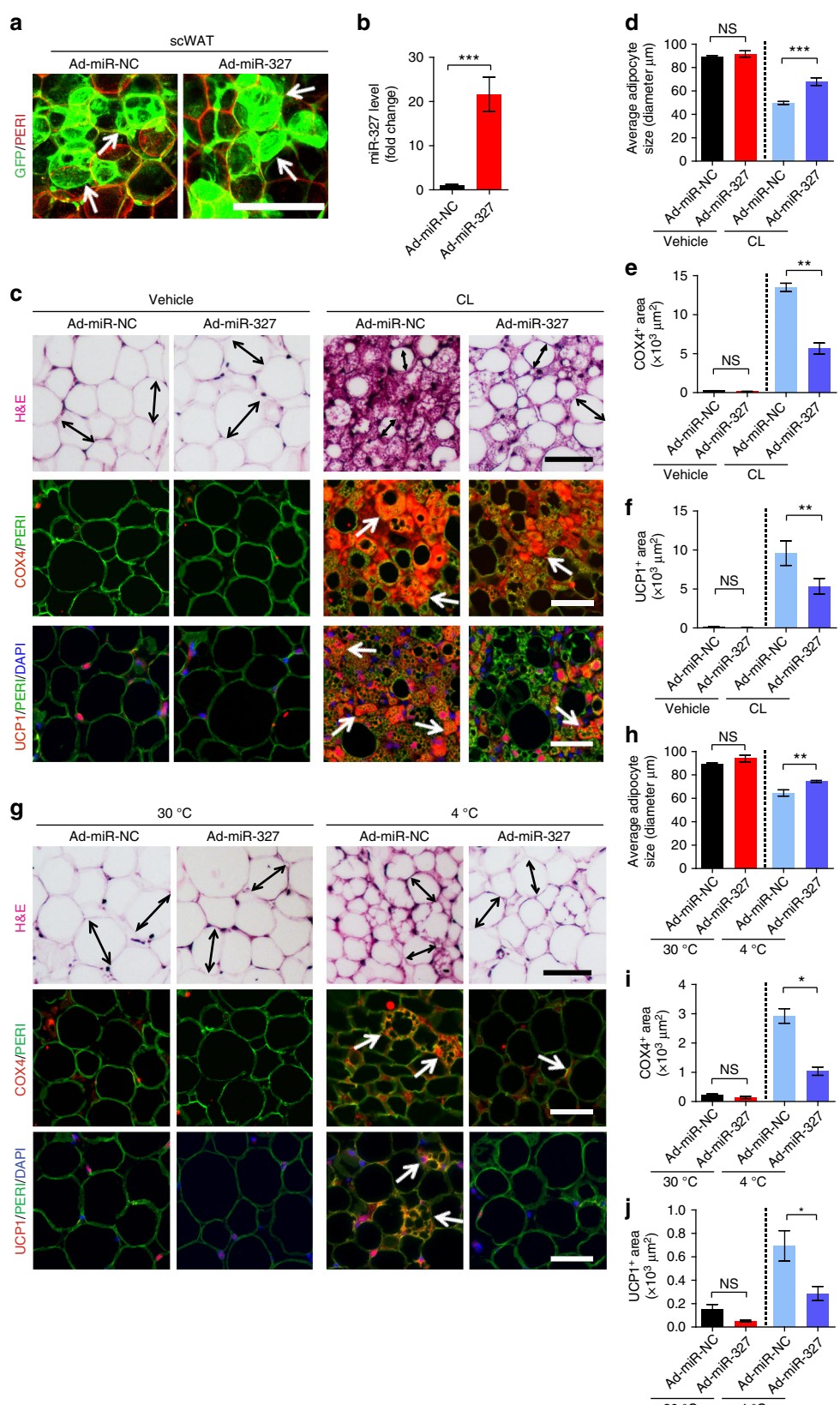

cold exposure further increased FGF10 levels (Fig. 9e, g, Supplementary Fig. 13a). Additionally, significant increases in UCP1 protein levels were detected in miR-327 inhibitor-treated WATs of 30 °C- and 4 °C-exposed mice (Fig. 9f, h, Supplementary Fig. 13b).

As systemic delivery of the miR-327 inhibitor might also affect iBAT, immunohistochemical analysis of adipocyte morphology and quantitative measurements of COX4, FGF10 and UCP1 were performed in miR-327 inhibitor treated mice. Unlike scWAT, systemic delivery of miR-327 inhibitor had no significant impact on iBAT (Supplementary Fig. 7). These findings further validate that the function of miR-327 is restricted to WAT browning but not BAT activation.

To determine the overall phenotype of mice treated with miR-327 inhibitor, body weight, food intake, tissue weight, glucose and insulin tolerance were measured (Fig. 9i, Supplementary Fig. 8a–f). Systemic treatment of mice with a miR-327 inhibitor did not affect total body weight or food intake under 30 or 4 °C conditions (Supplementary Fig. 8a, b). However, a trend of a slight decrease in tissue weight of scWAT and visWAT could be seen in these mice upon miR-327 inhibition (Fig. 9i, Supplementary Fig. 8c).

Measuring the metabolic rate, NE-induced oxygen consumption and core body temperature in miR-327 inhibitor treated mice exposed to 30 or 4 °C demonstrated an increase of total energy expenditure in mice lacking functional miR-327 (Fig. 9j–l, Supplementary Fig. 8g).

Our data highlight a functional impact of altered miR-327 levels on whole-body energy expenditure. All together, these gain- and loss-of-function studies suggest a major role of miR-327 in regulating white adipose tissue browning.

**Genetic depletion of *Fgf10* ablates a browning phenotype.** Knockout of the *Fgf10* gene in mice results in lethality at birth owing to a lack of lung development[23–25], which makes it impossible to study browning of adipose tissue in these mice. Therefore, *Fgf10* heterozygous (*Fgf10*[+/−]) mice were used to experimentally address the link between FGF10 levels and the phenotype observed in miR-327 treated mice. Deletion of only one *Fgf10* allele in heterozygous mice resulted in a significant reduction of FGF10 protein levels in scWAT, visWAT and iBAT (Supplementary Fig. 9a), which makes these mice a suitable model to validate the role of FGF10 as a target of miR-327 in an in vivo setting.

To validate the effect of *Fgf10* levels during miR-327 treatment, wild type (WT) and *Fgf10*[+/−] mice were treated with a miR-327 inhibitor at 30 °C. Although the average adipocyte size was slightly decreased in *Fgf10*[+/−] mice treated with miR-327 inhibitor, this effect was much less pronounced compared to the effect seen in WT mice. Furthermore, COX4 and UCP1 levels remained at very low levels after miR-327 inhibitor treatment in *Fgf10*[+/−] mice (Fig. 10a–d). Consistently, western blot analysis of miR-327 inhibitor treated *Fgf10*[+/−] mice demonstrated only a modest increase of FGF10 and UCP1 protein levels (Fig. 10e–h, Supplementary Fig. 14) compared to the increase observed in WT mice (Fig. 9e–h). Noticeably, the miR-327 inhibitor-induced

decrease of scWAT weight and increase in metabolic rate observed in WT mice were ablated in *Fgf10*[+/−] mice (Fig. 10i, j). Consistent with previous findings, *Fgf10*[+/−] mice did not exhibited any alterations in the miR-327 inhibitor-treated BAT (Supplementary Fig. 9b–h). Likewise, body weight, food intake, adipose depot weight, insulin sensitivity, and core body temperature remained unchanged in miR-327 inhibitor-treated vs. control vehicle-treated *Fgf10*[+/−] mice (Supplementary Fig. 9i–n).

## Discussion

In rodents and other mammals, WAT, especially scWAT, displays metabolic plasticity by either storing excessive energy or fueling energy expenditure[36, 37]. The switch from energy storage to thermogenesis is regulated by the sympathetic nervous system through the β3-adrenoceptor pathway[38, 39]. Environmental changes, such as cold exposure, drugs, diet, and disease can lead to WAT browning through sympathetic activation[8]. It is known that different WAT depots possess intrinsic properties that make them susceptible or resistant to browning. For example, scWAT in mice are susceptible to browning upon cold exposure, whereas visWAT is intrinsically resistant to become beige[9, 11, 14, 39]. Similar mechanisms of WAT browning also seem to exist in adult humans as substantial beige adipose depots have been detected in adult humans after exposure to cold or β3-adrenoceptor stimulation[4–6]. Along with the browning phenotype, beige WAT contains a markedly increased vascular density and an increased number of other SVF components[2, 3, 11–13]. Nonetheless, the functional impact of the SVF on controlling adipocyte metabolism remains a relatively unexplored area.

WAT browning is dependent on preadipocyte differentiation to beige adipocytes that express specific markers such as UCP1. Recent work from our laboratory and others supports the notion that preadipocyte differentiation is a key process for gaining beige cells in browning WAT[12, 28, 40]. Preadipocytes reside within the non-adipocyte fraction, suggesting that non-adipocytes, especially the SVF-derived cells, may play a crucial role in modulating the browning process[17]. In support of this view, our recent work shows that blocking SVF-preadipocyte differentiation markedly attenuates cold- and CL-induced WAT browning[12]. However, molecular players, signaling pathways and regulatory mechanisms underlying the differentiation of SVF-derived cells to beige adipocyte are still under investigation. Knowing that the SVF plays a central role in controlling WAT browning, we focus our present work on gaining in-depth mechanistic insights into regulatory and signaling molecules in the non-adipose fractions. Our findings suggest that the initial trigger of WAT browning might be located in the non-adipocyte compartment that supplies both preadipocytes for subsequent differentiation and signaling molecules that control beige adipocyte differentiation. However, interconversion of mature adipocytes from the WAT to beige phenotype has been proposed as another mechanism of browning[41]. Although it is unclear if differentiation or interconversion is the dominant process under different experimental settings, both possibilities clearly co-exist under physiological conditions.

**Fig. 7** Ad-miR-327 inhibits WAT browning. **a** Histological analysis of GFP[+] cells in scWAT transfected with a control adenovirus (Ad-miR-NC) or an adenovirus expressing miR-327 (Ad-miR-327). Perilipin (PERI) was used to identify adipose tissues. Arrows point to GFP[+] cells. **b** qPCR analysis of miR-327 expression in Ad-miR-327 compared to Ad-miR-NC. Sno-202 served as an internal control (n = 6 samples per group). **c, g** Histological analysis of adipocyte morphology (H&E), adipocytes (PERI), mitochondria (COX4) and uncoupling protein 1 (UCP1) in **c** 5-day CL-316243-treated scWAT compared to vehicle-treated control. **g** Two-week 4 °C-treated scWAT compared to 30 °C control. Double-headed arrows mark adipocyte diameter. Arrows point to respective positive signals. **d–f, h–j** Quantifications of adipocyte size and positive signals of COX4 and UCP1 in **d–f** CL-316243- and vehicle-, and **h–j** 30 °C- and 4 °C-treated scWATs (>30 adipocytes per field; n = 10 random fields; n = 6 mice per group). Immunodeficient NSG mice were used for all experiments in this figure. Scale bars, 100 μm. NS, not significant. *$P < 0.05$, **$P < 0.01$, and ***$P < 0.001$ by Student's t-test. Data presented as mean ± s.e.m.

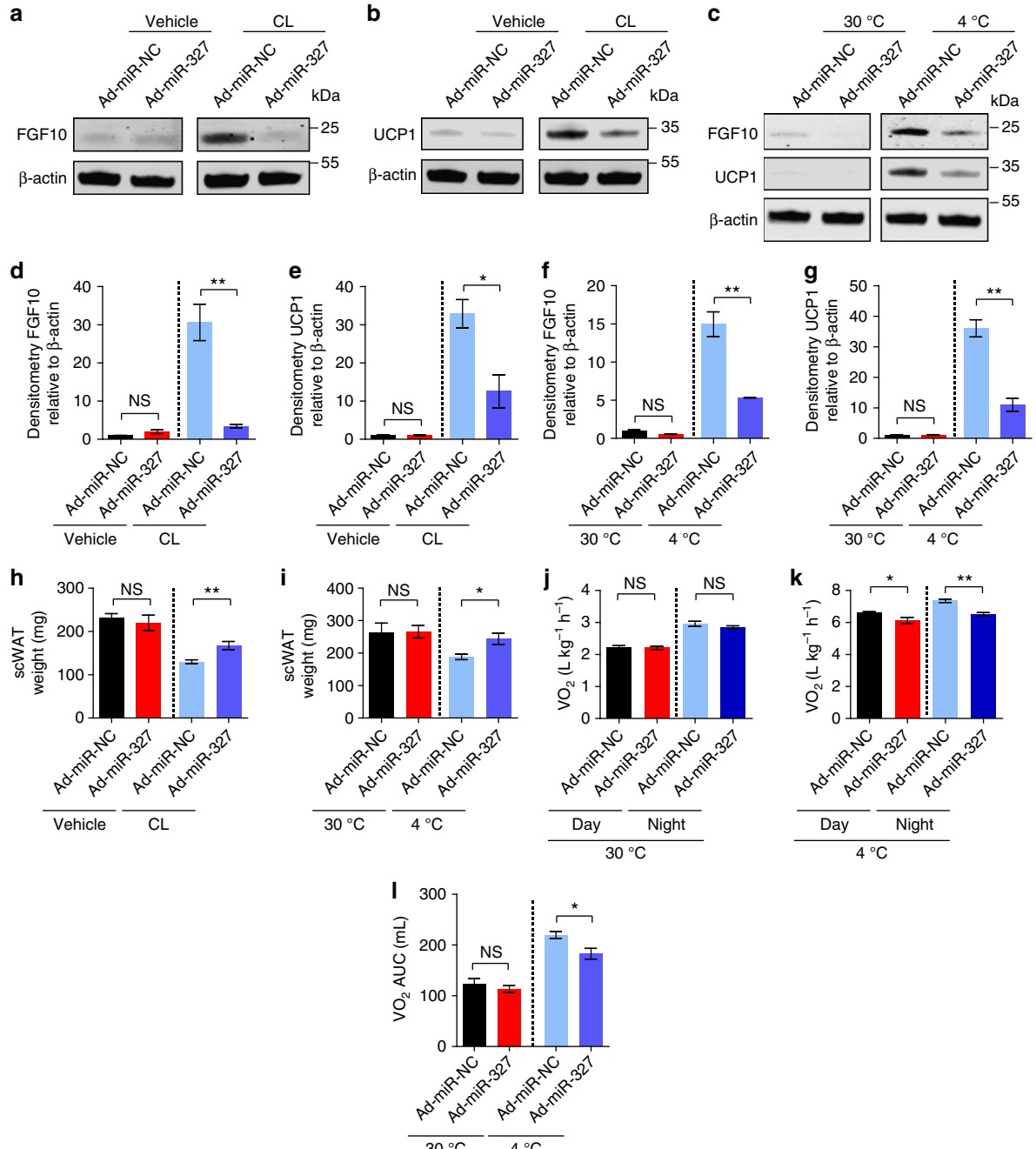

**Fig. 8** miR-327 overexpression leads to reduced oxygen consumption in cold-exposed mice. **a**–**g** Western immunoblot analysis and quantification of FGF10 and UCP1 of Ad-miR-NC- and Ad-miR-327- treated scWAT derived from 5-days vehicle or CL-316243-treated and 2-week 30 °C or 2-week 4 °C exposed NSG mice. FGF10 and UCP1 protein levels were quantified as densitometric signals and normalized to β-actin ($n = 4$ samples per group). **h**, **i** Depot weights of Ad-miR-NC- or Ad-miR-327-treated scWAT dissected from CL-316243-treated (5 days) and 4 °C- or 30 °C-treated (2 weeks) NSG mice ($n = 12$ mice per group) **j**, **k** Oxygen consumption of Ad-miR-NC- and Ad-miR-327-treated NSG mice under 30 °C or 4 °C for 2 weeks. Oxygen consumption was measured during the resting phase (Day) and active phase (Night) ($n = 6$ mice per group). **l** Norepinephrine (NE)-stimulated thermogenesis of Ad-miR-NC- and Ad-miR-327-treated mice after 2-week 30 °C or 4 °C exposure. Oxygen consumption is presented as area under curve (AUC) ($n = 4$ mice per group). Immunodeficient NSG mice were used for all experiments in this figure. kDa, kilodalton. NS, not significant. *$P < 0.05$, **$P < 0.01$, and ***$P < 0.001$ by Student's *t*-test. Data presented as mean ± s.e.m.

In this study, we show that cold exposure and CL treatment markedly increase the number of αSMA⁺ cells that are densely distributed throughout the browning adipose tissue. Although the origin of these cells warrants further investigation, they are likely to come from two possible sources. One possibility would be a redistribution of vessel-associated αSMA⁺ smooth muscle cells under browning conditions. Independent studies from other

laboratories also support the fact that perivascular cells serve as a reservoir for supplying adipocyte precursor cells[17, 42, 43]. A second possibility is the differentiation of existing stromal fibroblasts to αSMA⁺ myofibroblasts in browning WATs. The latter possibility suggests that differentiation of fibroblasts toward myofibroblasts is a prerequisite for further differentiation into beige cells. Thus, defining the signaling driving force committing

stromal cell differentiation would be essential to understand the mechanisms of WAT browning.

To decipher molecular mechanisms controlling beige cell differentiation, we focused our efforts on the non-adipocyte fractions where the mesenchymal stem cells and preadipocytes reside. Analysis of miRNAs in the SVF of the browning adipose tissue is

a novel approach. It allows us to reveal autocrine mechanism that are regulated on the post-transcriptional level and determine preadipocyte to beige adipocyte differentiation potentials. We show that FGF10 is a novel target for miR-327 and that the expression of miR-327 is altered during browning and FGF10 protein is profoundly increased in browning WATs. In non-

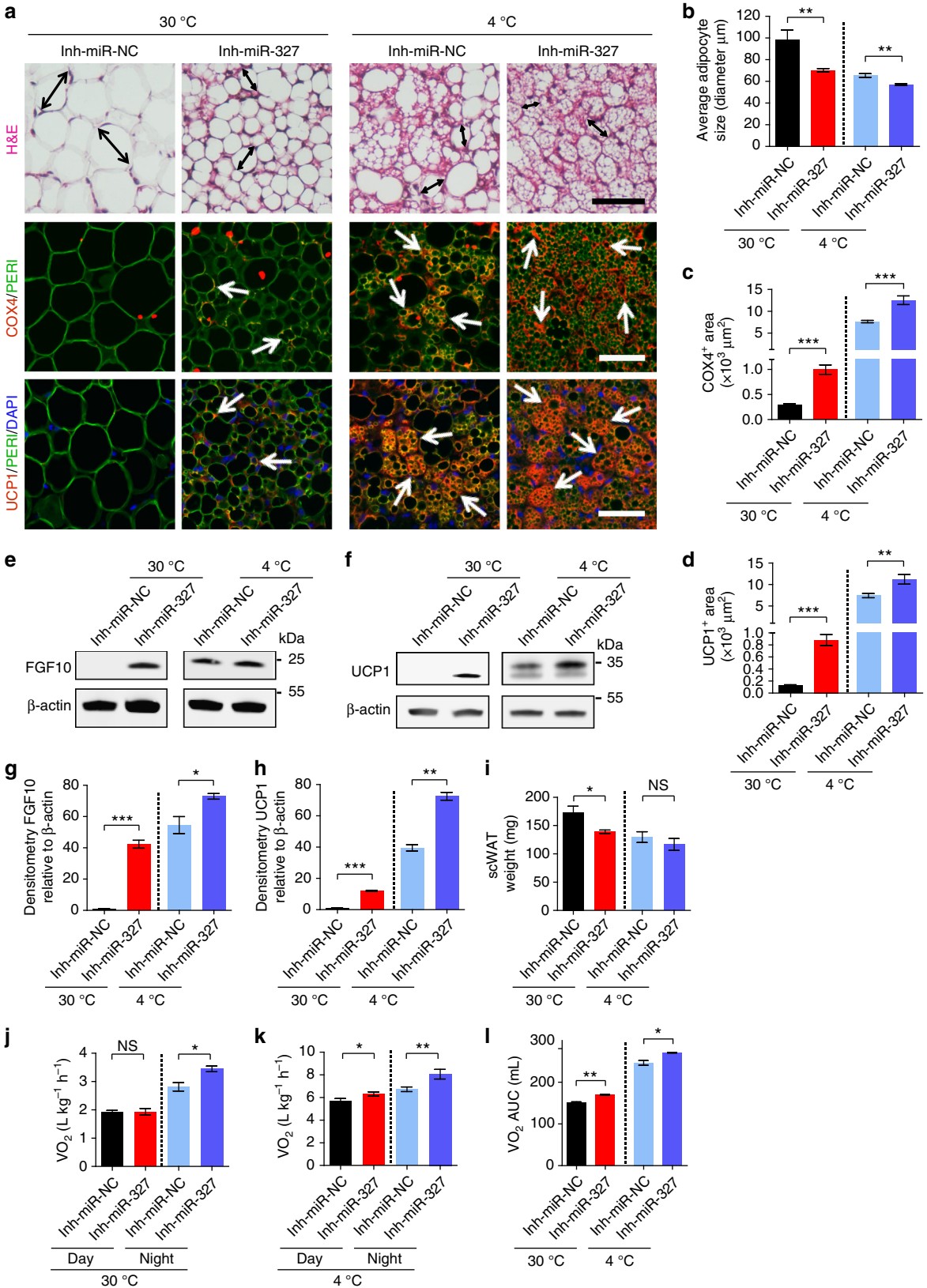

browning WAT, FGF10 exists in scWAT at low levels and was undetectable in visWAT. This finding suggests that FGF10 is involved in WAT browning and a low amount of FGF10 in visWAT might be linked to its resistance to browning. Although FGF10 protein is present in iBAT, its expression level remains unchanged under all experimental conditions. Thus, FGF10 is specifically responsible for WAT browning, but not iBAT activation. These data further imply the existence of differential mechanisms governing WAT browning and iBAT activation.

We also provide convincing evidence showing that FGF10 commits to preadipocyte differentiation, but not proliferation, through the FGFR2–Akt signaling pathway. Transfection of miR-327 alone can markedly suppress FGF10 levels and attenuates preadipocyte differentiation toward browning adipocytes. These findings suggest the existence of an autocrine regulatory loop at the post-transcriptional level, where SVF cells produce one or several factors that drive them into differentiation. A causational link of the FGF10–FGFR2 signaling was established by blocking FGFR functions. Inhibition of FGFR2 blocks preadipocyte differentiation. Furthermore, activation of the downstream signaling component Akt is essentially required for FGF10-stimulated preadipocyte differentiation. Autonomous mechanisms controlling beige adipocyte differentiation through a post-transcriptional autocrine signaling pathway has not been previously described. The miR-327–FGF10–FGFR2–Akt autocrine pathway defines a completely novel molecular mechanism for thermogenic regulation (Fig. 10k). Although we mainly claim an autocrine mechanism in the regulation of adipose browning in the present work, we cannot completely exclude the possibility of the existence of a paracrine loop[44]. Some of the mechanisms depicted here might be relevant not only to the browning process but also to overall adipocyte differentiation, as evidenced by some of the effects found systematically in the 3T3-L1 model.

Another interesting finding is that genetic deletion of only one allele the *Fgf10* gene resulted in marked reduction of FGF10 protein in *Fgf10*[+/−] mice, which attenuated the browning phenotype in scWAT after miR-327 inhibition. This genetic approach provides robust evidence that a causational link between miR-327 and FGF10 exists to modulate WAT browning. Again, BAT remains unaffected in *Fgf10*[+/−] mice with or without miR-327 inhibition. Therefore, both pharmacological and genetic loss-of-functions reproduce similar experimental data and strengthen our conclusions.

In an in vivo WAT browning model, we show that delivery of Ad-miR-327 effectively blocks cold-induced browning and UCP1 expression, validating the functional impact of this miRNA on thermogenesis. Delivery of a miR-327 inhibitor alone to 30 °C-exposed animals induces a browning phenotype without cold or drug stimulus. Furthermore, a miR-327 inhibitor exacerbates cold-induced WAT browning and thermogenic activity. Consequently, systemic delivery of the miR-327 inhibitor significantly increases whole-body metabolism under both thermoneutral and cold conditions. This discovery is a proof-of-concept that microRNAs can be used for treatment of obesity, T2DM and other metabolic disorders. If these preclinical findings can be successfully translated into clinical practice, miRNAs targeting FGF10 would offer an effective novel therapeutic approach. Thus, our results provide mechanistic insights and pave a novel avenue for treatment of obesity and metabolic disorders.

## Methods

**Animal models.** Six-week-old immunocompetent female C57BL/6 (Charles River) and immunodeficient female NOD.Cg-*Prkdc*[scid] *Il2rg*[tm1Wjl]/SzJ (NSG) mice were obtained from the Department of Microbiology, Tumor and Cell Biology, Karolinska Institute, Stockholm, Sweden. Female *Fgf10*[+/−] C57BL/6 mice[45] were bred at Tokushima University. All experimental procedures related to mouse studies in Sweden were approved by the Northern Stockholm Animal Ethical Committee. All animal care and experiments performed on *Fgf10*[+/−] mice were carried out in accordance with the guidelines for animal experiments of Tokushima University, and were approved by the Ethics Committee of Tokushima University for Animal Research.

**In vivo drug treatment.** Eight-week-old-female mice were transferred to thermoneutral conditions (30 °C) 2 weeks prior to experimentation, followed by exposure to 4 °C or continual exposure to 30 °C for time spans indicated in the figure legends. CL-316243 disodium salt (Cat. No. 1499, Tocris Biosciences) was administered systemically by daily intraperitoneal injection at a dose of 1 mg kg[−1] in 0.1 mL PBS. Custom miRCURY in vivo LNA inhibitor probes (Exiqon) were used against mmu-miR-327 and a LNA inhibitor control served as negative control. Inhibitors at the concentration of 8 mg kg[−1] were subcutaneously injected once per week. An adenovirus overexpressing mmu-miR-327 (Ad-EF1a-mmu-miR-327-eGFP; 1 × 10[9] PFU; Vector Biosystems) was injected into scWAT and visWAT. An Ad-EF1a-miR155-Ctrl-miR-eGFP vector served as vehicle control. NSG mice were used for all experiments involving the in vivo delivery of adenovirus. Mice were pre-treated with adenoviruses or inhibitors 1 week prior to exposure to 4 or 30 °C.

**Isolation and fractionation of adipose tissues.** Cellular fractions containing primary adipocytes and SVF cells were isolated from 4- to 8-week-old C57BL/6 female mice by a standard collagenase-based digestion method[46]. Briefly, freshly isolated adipose tissues were minced and digested at 37 °C in digestion medium (0.2% collagenase II, 5% FBS in DMEM) for 30 min with shaking. Mature adipocyte fractions and stromal-vascular fractions were separated by incubation on ice for 10 min, followed by centrifugation at 300×g for 10 min. The adipocyte fraction was removed and the SVF was filtered through a 100-μm mesh, followed by washing twice with PBS and filtering through a 40-μm mesh. Mature adipocyte fractions and SVFs were directly used for RNA isolation, plated on collagen-coated plates for in vitro studies or were used for magnetic activated cell sorting (MACS).

**Cell differentiation assay.** Mouse 3T3-L1 embryonic preadipocyte cells (ATCC) were maintained in DMEM supplemented with 10% FBS. Primary SVF cells were maintained in DMEM/F-12, GlutaMAX (Thermo Fisher Scientific) containing 15% FBS. PureCol (Advanced BioMatrix) coated plates were used for all differentiation assays. 3T3-L1 preadipocyte differentiation was induced by the addition of 0.5 mM IBMX, 0.1 μM dexamethasone and 0.01–1 μg mL[−1] insulin for 2 days followed by treatment with 0.01–1 μg mL[−1] insulin for 4–6 days. Primary mouse cells were differentiated with 10% FBS in DMEM/F-12 containing 0.5 mM IBMX, 0.1 μM dexamethasone, 1 μM rosiglitazone, and 1 μg mL[−1] insulin for 4 days followed by treatment with 1 μg mL[−1] insulin for 4–6 days. Additional drug treatment was performed 24 h prior to initiation of differentiation and throughout the differentiation process. FGF10 (10 ng mL[−1], Peprotech) or Akti-1/2 (10 nM, Abcam) were used for the treatment. Isoproterenol at a concentration of 10 μM was added 4 h prior to RNA extraction.

**Fig. 9** Exacerbation of WAT browning by a miR-327 inhibitor. **a** Histological analysis of adipocyte morphology (H&E), adipocytes (PERI), mitochondria (COX4) and uncoupling protein 1 (UCP1) in Inh-miR-NC- and Inh-miR-327-treated 2-week 30 °C and 2-week 4 °C-exposed scWAT. Double-headed arrows mark adipocyte diameter. Arrows point to respective positive signals. **b–d** Quantifications of adipocyte size and positive signals of COX4 and UCP1 in Inh-miR-NC- and Inh-miR-327-treated 2-week 30 °C and 2-week 4 °C-exposed scWAT (>30 adipocytes per field; n = 10 random fields; n = 4 mice per group). **e–h** Western immunoblot analysis and quantification of FGF10 and UCP1 in Inh-miR-NC- and Inh-miR-327-treated 2-week 30 °C and 2-week 4 °C-exposed scWAT. FGF10 and UCP1 protein levels were quantified as densitometric signals and normalized to β-actin (n = 4 samples per group). **i** scWAT weight of C57BL/6 mice receiving Inh-miR-NC or Inh-miR-327 treatment under 2-week 30 °C or 4 °C exposure (n = 6 mice per group). **j, k** Oxygen consumption of Inh-miR-NC- and Inh-miR-327-treated C57BL/6 mice after 2-week 30 °C or 4 °C exposure (n = 6 mice per group). **l** Norepinephrine-stimulated thermogenesis of Inh-miR-NC- and Inh-miR-327-treated C57BL/6 mice after 2-week 30 °C and 2-week 4 °C exposure. Oxygen consumption is presented as area under curve (AUC) (n = 4 mice per group). Scale bars, 100 μm. kDa, kilodalton. NS, not significant. *P < 0.05, **P < 0.01, and ***P < 0.001 by Student's t-test. Data presented as mean ± s.e.m.

**In vitro treatment with siRNA, miRNA mimics, and miRNA inhibitors.**
Monolayers of 3T3-L1 or primary SVF cells at about 90% confluency were transfected with siRNAs using the DharmaFECT siRNA transfection protocol (Dharmacon). ON-TARGETplus SMARTpool siRNAs against *Fgfr2* and *Fgf10*, miRIDIAN mmu-miR-327 mimic, mmu-miR-327 miRCURY LNA inhibitor, and their respective negative controls were used for transfection at a final concentration of 25 nM using DharmaFECT transfection reagent 1 (GE Healthcare) diluted in OPTI-MEM reduced serum medium (Thermo Fisher Scientific Inc.). qPCR analysis was used to quantify transfection efficiency. FGF10 protein levels in the lysates

and conditioned media were quantified by an ELISA (Cat. No. M1−65673, Antibodies-online GmbH).

**Dual-luciferase reporter assay.** A psiCHECK−2 vector and a Dual-Luciferase Reporter Assay System (C8021, E1910, Promega) were used for target validation. Two ~700 bp long sequences of the 3′UTR of *Fgf10* mRNA that contained either miR-327 binding site 1 (bs1) or bs2 were separately cloned into a psiCHECK-2 vector. HEK293T human kidney cells (ATCC) were transfected with a

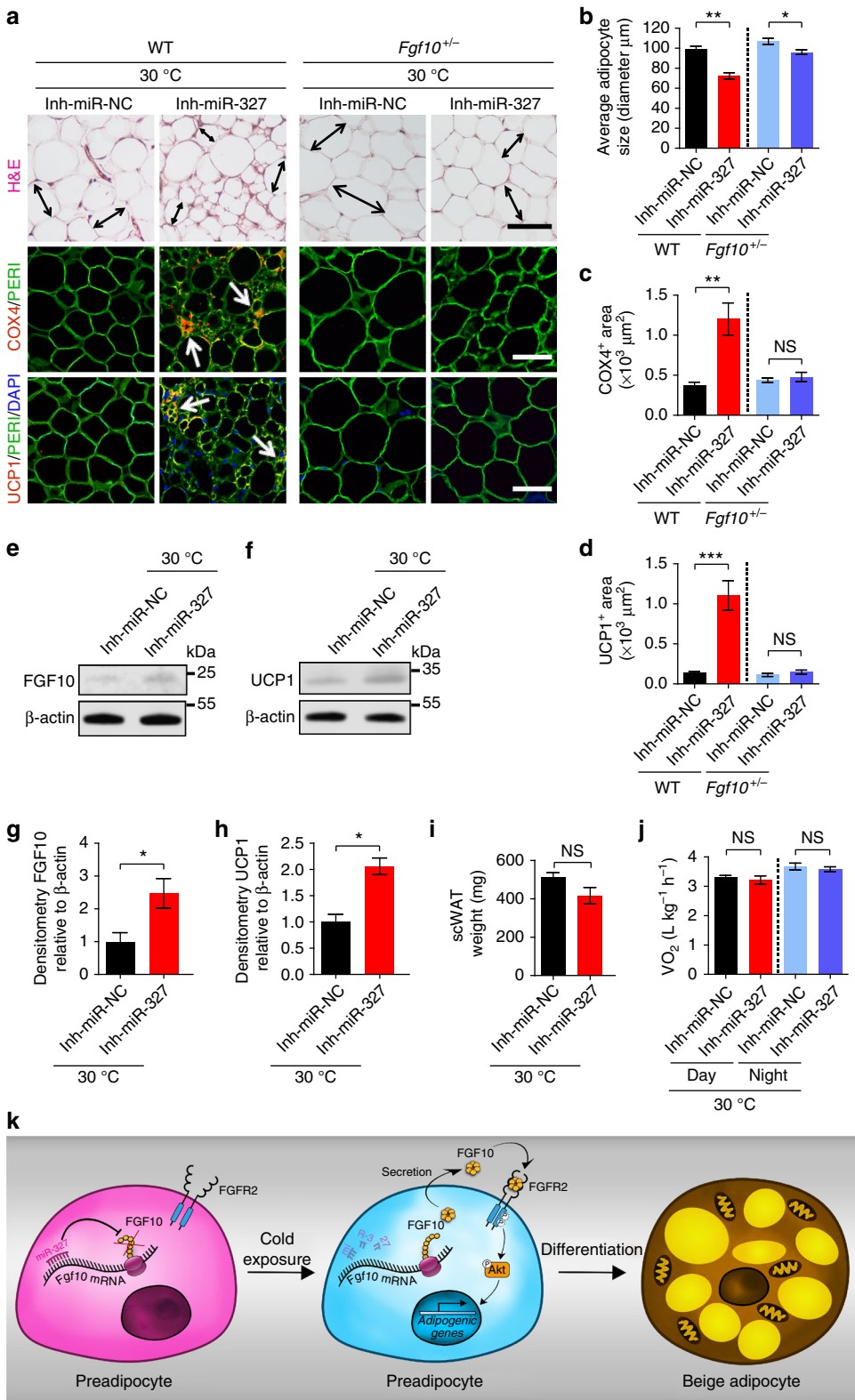

transfection medium containing the recombinant plasmids (1 µg µL$^{-1}$ DNA) mixed with Polybrene transfection reagent (1 mg mL$^{-1}$, Sigma-Aldrich) in OPTI-MEM medium (Thermo Fisher Scientific Inc.) for 16 h. Cells were further transfected with miR-327 mimics using DharmaFECT siRNA transfection reagent. Renilla and Firefly luciferase signals were measured 48 h after the initial transfection using the Dual-Luciferase Reporter Assay Protocol (Promega) and a Sirius L Tube Luminometer (Titertek-Berthold). To mutate the binding sites of miR-327 in the 3′UTR of Fgf10 mRNA, PCR amplification with primers containing 5 mismatches in the miR-327 bs1 or bs2 were used and the mutated recombinant plasmids were inserted into the psiCHECK-2 vector for further luciferase quantification. All primers used are listed in Supplementary Table 1.

**RNA isolation and qPCR.** A QIAzol-Chloroform based protocol was used for the extraction of RNA from tissue samples. Briefly, QIAzol lysis reagent (Qiagen) was added to cells or minced tissues followed by homogenization with a VDI 12 homogenizer (VWR). The samples were incubated on ice for 10 min before adding chloroform followed by centrifugation at 7000×g for 20 min. RNA-containing supernatants were mixed with 100% ethanol and subsequently applied to RNA extraction columns provided in the RNA extraction kit. All kits were purchased from Fisher Scientific (Thermo Fisher Scientific Inc.) if not indicated otherwise and all procedures were performed according to the manufacturer's instructions. Total RNA concentrations were measured with a NanoDrop2000 UV–Vis spectrophotometer and an equal amount of RNA from each sample was used for cDNA synthesis using a RevertAid cDNA synthesis kit. The cDNAs were subsequently used for qPCR analysis with Power SYBR Green Master Mix using the StepOnePlus Real-Time PCR System. For the detection of miRNA levels, a miRNeasy Mini Kit (Qiagen) was used together with the TaqMan MicroRNA Reverse Transcription Kit, the miRNA assay against miR-327 and the TaqMan Universal Master Mix. The amplified products were separated on GelRed (Biotium) supplemented agarose gels and were detected in a Gel Doc XR$^{+}$ (Bio-Rad Laboratories) to validate product size and primer specificity. All samples were ran in duplicates and qPCR data is presented as relative quantification with Actb or sno-202 serving as internal control for mRNA or miRNA expression analysis respectively. All primers used are listed in Supplementary Table 1.

**MACS sorting.** Different cellular fractions of adipose tissues were sorted using magnetic MicroBeads and a OctoMACS Separator (Cat. No. 130-042-108, Miltenyi Biotec) according to the manufacturers' protocols. In brief, SVFs derived from adipose tissues were filtered through a 40-µm-filter and counted. Cells were centrifuged at 300×g for 10 min and resuspended in 90 µL of a MACS buffer (PBS, pH 7.2, 0.5% BSA and 2 mM EDTA). CD45 microbeads (Cat. No. 130-052-301, Miltenyi Biotec) in 10 µL per 10$^7$ cells were added and samples were incubated on ice for 15 min. Cell were washed by adding 1 mL of MACS buffer, centrifuged and resuspended in 500 µL MACS buffer. For cell separation, MS columns (130-042-201, Miltenyi Biotec) were placed on the OctoMACS separator and rinsed with 500 µL MACS buffer, followed by applying cell suspension. The flow through was collected as negative fraction followed by three washing steps. To collect positive cell fractions, MS columns were removed from the magnetic holder and flushed with 1 mL MACS buffer. After depletion of CD45$^{+}$ cells, the negative fraction was incubated with either CD31 (Cat. No. 130-097-418, Miltenyi Biotec) or PDGFR-α (Cat. No. 130-101-502, Miltenyi Biotec) microbeads and sorted as described above. Total RNAs were isolated from the collected fractions using the above described QIAzol-Chloroform-based protocol.

**Immunoblotting.** Protein lysate was prepared using a Triton X-100 based cell lysis buffer containing proteinase inhibitors (Sigma-Aldrich) and phosphatase inhibitors (Cell Signaling Technology). For protein separation, a standard molecular weight marker and an equal amount of protein from each sample were loaded on a NuPAGE Novex 4–12% Bis-Tris SDS–PAGE gel (Thermo Fisher Scientific Inc.), followed by wet transfer onto a 0.45 µm nitrocellulose membrane which was subsequently blocked with a 5% BSA in PBST solution for 30 min. Overnight incubation with a specific primary antibody at 4 °C was followed by incubation with a secondary antibody conjugated with IRDye 680RD or IRDye 800CW (LI-COR Biosciences) for 1 h at RT. Target proteins were detected using the Odyssey CLx Imaging System (LI-COR Biosciences). β-actin was used as a control for all western

blots. Primary antibodies used: β-actin (1:5000, Cat. No. 3700), Akt (1:1000, Cat. No. 9272), phospho-Akt (1:1000, Cat. No. 4051), Erk (1:1000, Cat. No. 4695), phospho-Erk (1:1000, Cat. No. 9101), phospho-FRS2α (1:1000, Cat. No. 3864) from Cell Signaling Technology, and FGF10 (1:2000, Cat. No. ABN44) from Merck Millipore.

**ELISA assay.** Primary and 3T3-L1 preadipocytes cultured in monolayer were counted, washed twice with PBS, and lysed with a lysis buffer (Cat. No. 3228, Sigma), followed by 20-min centrifugation to remove cellular debris. Conditioned medium was collected at 72 h after drug treatment and was centrifuged to remove cellular debris before use. Serum samples were prepared from whole blood using standard serum separation tubes (Cat. No. 365967, BD Biosciences). To quantify FGF10 protein levels, a FGF10 ELISA kit (Cat. No. M1-65673, Antibodies-online GmbH) was used. Serum insulin levels were detected using a Mouse Insulin ELISA Detection Kit (Cat. No. 10-1247-01, Mercodia AB). Both kits were used according to the companies' instructions and the absorbance values were detected at 450 nm using a microplate reader.

**Oral glucose tolerance tests.** Two- to four-month-old-female mice were fasted for 6 h during the light phase with free access to water. Tail vein blood was used to measure glucose levels using the Accu-Chek Aviva glucometer and strips (Roche Diagnostics). Glucose levels were measured twice for every time point (0, 15, 30, 60, and 120 min) after oral administration of 15 mg of glucose (Cat. No. G8270, Sigma-Aldrich) per g mouse dissolved in NaCl.

**HOMA-IR.** Homeostatic model assessment of insulin resistance (HOMA-IR) was employed to measure the fasting insulin levels in serum samples using a mouse Insulin ELISA kit. Fasting blood glucose levels were measured using the Aviva glucometer before killing animals after 6 h starvation. The values were calculated using the following formula: Fasting glucose (mg dL$^{-1}$) × fasting insulin (mU L$^{-1}$)/405.

**Proliferation assay.** A non-radioactive cell proliferation assay (G5421, Promega Corporation) was used according to the manufacturers protocol (G5421 protocol, Part# TB169). In brief, the viability of cells was measured at 72 h after in vitro treatment or at the indicated time points by adding 20 µL of combined MTS/PMS solution to each well of a 96-well plate, followed by incubation at 37 °C for 1–4 h. Absorbance at 490 nm was measured by a microplate reader.

**Microarray analysis.** Genome-wide Affymetrix mouse gene 2.0 ST microarray (deposited in Gene Expression Omnibus (GEO) under accession code GSE55934) and Agilent mouse miRNA microarray (Cat. No. G4872A-046065, Agilent Technologies) was performed on the SVF of visWAT of 8-week-old C57BL/6 mice treated with CL-316243 or Vehicle (PBS) for 3 days. Isolated miRNA and total RNA was used for array analysis by Shanghai Biotechnology Corporation. Normalization and analysis for differentially expressed genes were performed using robust multi-array analysis and significance analysis of microarrays (SAM) via R statistical software packages "oligo" and "samr". Heatmaps were presented for significantly upregulated and downregulated genes using Microsoft Excel 2010. MiRBase was used for obtaining miRNA sequences and microRNA.org, PicTar, TargetScan, and miRDB were used for the identification of potential miRNA targets.

**H&E staining and immunohistochemistry.** Paraffin-embedded adipose tissues were cut into 5–10 µm thick sections, adhered onto superfrost glass slides, deparaffinized (Tissue-Clear, Sakura Finetek), and rehydrated by decreasing ethanol concentrations. Tissue-embedded slides were stained with hematoxylin and eosin (H&E), followed by dehydration and mounting with PERTEX (HistoLab Products AB). For immunohistochemistry, rehydrated tissues were washed with PBS, boiled in a low-pH unmasking solution (ZB0611, Vector Laboratories), and blocked with 4% serum before overnight incubation at 4 °C with primary antibodies against COX4 (1:300, Cat. No. GTX114330, GeneTex), UCP1 (1:300, Cat. No. ab10983, Abcam), Perilipin (1:400, Cat. No. 20R-PP004, Fitzgerald Industries), and αSMA (1:300, Cat. No. M085101–2, Agilent Technologies). Fluorescent-labeled secondary antibodies were purchased from the Thermo Fisher Scientific Inc. and were used at a dilution of 1:400. The slides were thoroughly washed and

**Fig. 10** Essential role of FGF10 in mediating Inh-miR-327-induced WAT browning. **a** Histological analysis of adipocyte morphology (H&E), adipocytes (PERI), mitochondria (COX4), and uncoupling protein (UCP1) in scWAT isolated from Inh-miR-NC- and Inh-miR-327-treated 2-week-30 °C-exposed WT and Fgf10$^{+/-}$ mice. Double-headed arrows mark the adipocyte diameter. Arrows point to respective positive signals. **b–d** Quantifications of scWAT adipocyte size and positive signals of COX4 and UCP1 in Inh-miR-NC- and Inh-miR-327-treated 2-week-30 °C-exposed WT and Fgf10$^{+/-}$ mice (>30 adipocytes per field; n = 10 random fields). **e–h** Western immunoblot analysis and quantification of FGF10 and UCP1 in Inh-miR-NC- and Inh-miR-327-treated scWAT of 2-week 30 °C-exposed Fgf10$^{+/-}$ mice. FGF10 and UCP1 protein levels were quantified as densitometric signals and normalized to β-actin (n = 4 samples per group). **i** scWAT weight of Fgf10$^{+/-}$ mice treated with Inh-miR-NC or Inh-miR-327 under 2-week 30 °C (n = 4 samples per group). **j** Oxygen consumption of Inh-miR-NC- and Inh-miR-327-treated Fgf10$^{+/-}$ mice after 2-week 30 °C exposure (n = 6 measurements per group). **k** Graphical display of the interplay between miR-327 and FGF10 in adipocyte differentiation and browning. Scale bars, 100 µm. kDa, kilodalton. NS, not significant. *P < 0.05, **P < 0.01, and ***P < 0.001 by Student's t-test. Data presented as mean ± s.e.m.

mounted with Vectashield containing DAPI Cat. No. V-1200, Vector Laboratories), followed by analysis under a fluorescence microscope (LSM510, Zeiss). Ki67 (1:300, Cat. No. 14-5698-82, eBioscience) was used for cell immunohistochemical staining. Cells were fixed for 30 min in 4% PFA, washed twice with PBS-Triton X, and incubated with primary and secondary antibody. Positive signals were detected as described above.

**Oil Red O staining**. Lipid Oil Red O staining was performed according to the BioVision manual (BioVision). In brief, cells were fixed with 4% PFA for 30 min, washed with 60% isopropanol, and stained at 37 °C with Oil Red O (Cat. No. O1391, Sigma-Aldrich) solution for 15 min. Nuclei were counterstained with hematoxylin for 10 min, mounted with glycerol, and examined under a light microscope.

**Whole-mount staining**. Whole-mount tissues were stained according to our previously published methods[47–49]. Briefly, tissues were fixed at 4 °C with 4% PFA in PBS overnight, washed with PBS, and cut into small pieces. Tissues were digested with proteinase K (20 mM) and permeabilized with 100% methanol. Samples were blocked overnight with a PBS-0.1% Triton X-100 solution containing 3% milk, and incubated at 4 °C overnight with primary antibodies against CD31 (1:100, Cat. No. AF3628, R&D) and Perilipin (1:200, Cat. No. 20R-PP004, Fitzgerald Industries). After washing with PBS, samples were blocked with 3% milk, and incubated at RT with fluorescent-conjugated secondary antibodies (1:300, Thermo Fisher Scientific Inc.) for 2 h. After rigorous rinsing, samples were mounted with a Vectashield mounting medium (Cat. No. H-1000, Vector Laboratories). Fluorescent signals were examined under a confocal microscope (LSM510 Confocal, Zeiss).

**Indirect calorimetry**. Metabolic rate and NE-induced thermogenesis was quantified by measuring oxygen consumption of each 2- to 4-month-old-female mouse over a defined period of time using the Oxymax CLAMS-HC System (Columbus Instruments). An oxygen sensor was heated up for at least 6 h prior to calibration with the reference gases—100% nitrogen gas and a mixture of 20.5% $O_2$ and 0.5% $CO_2$. For measuring basal metabolism, mice were transferred to Oxymax chambers placed at different temperatures. These mice were kept for 24 h before recording started, followed by a 48–72 h recording period. For NE-induced thermogenesis the Oxymax machine was kept at 30 °C. Metabolic rates were measured as baselines for 24 h prior to subcutaneous NE injection (1 mg NE bitartrate $kg^{-1}$). The area under curve (AUC) of the volume of oxygen ($VO_2$) consumed was calculated for a period of 30 min prior to NE injection until 140 min after injection.

**Core body temperature**. Mouse core body temperature was measured in 2- to 4-month-old-female mice with a MicroTherma 2T thermometer and an animal rectal probe (Cat. No. TW2 and RET-3, Agntho's AB) by placing the probe into the rectum of each mouse while exposed to various temperatures.

**Statistical analysis**. Quantitative values are presented as mean determinants (±s.e.m.). For two-group comparisons, two-sided unpaired Student's $t$-test was performed. For RNA array analyses, $P$- and $Q$-values were calculated. Excel 2010 and GraphPad Prism 5 software were used for graphical representations. All groups were analyzed and evaluated to be appropriate for assumption of normal distribution and variance was similar between groups. For data sets with <5 samples per group, representative pictures or individual data points are shown. Sample size for animal studies was determined empirically based on experience through previous studies. Animals were separated into different groups to optimize a similar body weight for all groups before experiments were started. No animals were excluded from statistical analysis and the investigators were not blinded in the study. $P$-values of *$P < 0.05$ were considered statistically significant.

**Data availability**. Genome-wide Affymetrix mouse gene 2.0 ST microarray data were deposited into Gene Expression Omnibus (GEO) under accession code GSE55934. The Agilent miRNA microarray data were deposited into GEO under accession code GSE105226. All relevant data are available from the authors.

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

## Acknowledgements

Y.C.'s laboratory is supported by research grants from the Swedish Research Council, the Swedish Cancer Foundation, the Karolinska Institute Foundation, the Torsten Soderberg Foundation, the European Research Council (ERC) advanced grant ANGIOFAT (Project no. 250021), the NOVO Nordisk Foundation advanced grant, and the Royal Alice Wallenberg Foundation. C.F. is supported by KID funding.

## Author contributions

C.F. generated ideas and wrote the manuscript. C.F. performed most of the experiments and Y.C. and C.F. designed the figures. T.S., A.Y., M.N., P.A., Y.Y., J.H., M.M. and J.Z. participated in the experimentation. S.L., S.O., Y.Z., Y.W., Y.G., F.C., X.L., Y.L. and N.S. provided experimental reagents and participated in the discussions.

## Additional information

**Competing interests:** The authors declare no competing financial interests.

