## [Peer Review File · Nature Communications]

Reviewers' Comments:

Reviewer #1:

Remarks to the Author:

General comment:

The manuscript by Fischer et al. reports the identification of miR-327 as a regulatory factor in the control of browning of adipose tissues. It claims that the effects of miR-327 (a negative regulator) occur via the induction of FGF10 which acts in an autocrine manner. The study provide interesting data and add one more actor to the growing list of small regulatory RNAs involved in the control of brown/beige adipose biology.

Weaknesses of the manuscript are: the lack of identification of the cell types accounting for some of the phenomena observed in the rather heterogeneous SVF fraction of adipose tissue, lack of measurement of the extracellular release of FGF10, necessity of a stronger demonstration that it is effectively FGF10 the major mediator of miR-327 effects, and necessity of a more thorough assessment of the systemic alterations in the in vivo models reported in the last part of the manuscript.

Specific points:

- Why data obtained in Fig 2 prioritized the use of visceral WAT, where the browning process is much weaker than scWAT?
- The Results section in page 7 refers systematically to data shown in Fig2 as tissues (scWAT, vsWAT, iBAT) whereas data correspond to SVF isolated from tissues. The paragraphs should be written in a way that it states more explicitly that they correspond to SVF from tissues. By the way, providing data of the actors of the system proposed (miR-327, FGF10, FGFR2) in the tissue/mature fraction would be of interest to assess the specificity of the effects found in non-adipose versus mature adipose fractions, even if basal levels could be lower.
- Again, for consistency in the presentation of data: Fig2 should indicate explicitly when data correspond to mRNA (panel j to m). Otherwise, in Fig 3, panels a and b appear to correspond to data on whole tissue, not SVF. What's the reason for not determining the FGF10 protein in SVF as in Fig 2? The interpretation of data may be rather affected. What about the influence/ behavior of the mature cell fraction, excluded in the mRNA measurements in Fig 2 but present in the protein measurements in Fig 3?
- Claims for paracrine effects of FGF10 should include measurements of the FGF10 protein in the extracellular compartment in several experimental settings. The absence of these data is a weakness of the manuscript as it claims explicitly for autocrine actions.
- The experiment shown in Fig 4h is essential to demonstrate that the effect of miR-327 down-regulation leading to promotion of adipogenic differentiation and the effect of FGF10 promoting also differentiation were two linked phenomena, and not two independent processes occurring in parallel. The statements that FGF10 rescues miR-327 driven impairment in differentiation should be in some way toned down according to Fig 4h. In fact, the fold-induction in differentiation elicited by FGF10 in the absence of miR-327 (the two NC panels) appears not so different from that elicited in the presence of miR-327 (the two miR-327 panels). The conclusive sentence in this regard in lines 244, 245 and the heading in 224 should be toned down accordingly. Moreover, models using loss-of-function for FGF10 would be needed to conclusively rule out that miR-327 is not acting through pathways other than FGF10.
- Also in relation to the experiments depicted in Fig 4, in fact the data provided in a-j panels concern general adipogenesis (3T3-L1 are not cell models of beigeing). seems thaDoes it mean that miR-327 affects adipogenic differentiation regardless of whether this is "white" differentiation or "beige" differentiation? If cell models and in vivo models were used to check for white adipogenic differentiation instead of the models of browning used across the paper and they were challenged for miR-327 manipulation, would conclusions be the same?
- Proper assessment of the data obtained in the experiment shown in panels m and n of Fig 4 (transcript levels in the presence of isoproterenol) should include the basals, showing data only "in the presence of Isoproterenol", without the basals of untreated cells, makes no possible a precise

interpretation of the data.

- Why no data are provided on the FGF10/FGFR2 system in the in vivo experiments shown in Fig 5? This is needed for consistency in the core scientific message of the manuscript.
- In relation to the experiments depicted in the last part of Fig 6., considering that the administration of the miR-327 inhibitor is not adipose-driven, systemic data in these mice in addition to oxygen consumption should be provided to properly assess the direct versus indirect effects leading to the differences in WAT browning observed. Moreover, iBAT behavior (as responsible for most adaptive thermogenesis activity in mice) in response to the inhibitor should be shown. Finally, the changes of NE-induced oxygen consumption in mice should be shown in more detail, the data on NE-induced oxygen consumption should be accompanied by the oxygen consumption data in basal, before NE-treatment, in the miR-327 gain- and loss-of-function manipulations in mice. The significant but rather mild effects found between groups require a precise assessment of these data.

Other specific points are:

Some of the statements in Introduction about the physiological role of WAT browning (lines 68-72) should be toned down, the relative physiological role of WAT browning is rather controversial (see Nedergaard et al. *Cell Metab.* 2014 20:396-407, or Nedergaard et al. *Biochim Biophys Acta.* 2013 1831:943-9).

To be strict, mention on how miRNAs control protein production should include inhibition of translation, in addition to mRNA down-regulation (lines 85-87)

The use of a highly artificial system for the assessment of the targeting of miRNA327 upon FGF10 mRNA (including the use of HEK297 cells, of renal origin) should lead to tone down the conclusion in this regard, as the whole rationale of the manuscript concerns the pre-adipocyte/adipocyte context.

Fig 2. Differences in gene expression by comparing SVFs can be affected by differential cell composition (biological) but also to technical aspects of preparation by using as sources for isolation tissues with rather different in structure (because of being control tissues or CL-treated tissues). Please comment.

Representation of phosphorylation of Akt and ERK is unclear. The ratio between phosphorylated and total protein should be shown. Otherwise, Page 10 states no significant differences in P-ERK, but a $P < 0.001$ difference (minor, but apparently highly significant) is shown in Fig3 q (?). Providing the phosphorylated/non-phosphorylated ratios may help for clarity.

The heading of the last Fig. should be changed, as it includes not only data with the miR-327 inhibitor but also with the adenoviruses used in the previous Figure. Perhaps the oxygen consumption data in the adenovirus-treated mouse model should be moved to the previous Figure.

Write in full the number after CL in the Figure legends

In Fig2, indicate explicitly when data correspond to mRNA (panel j to m).

Please, provide details on sex of mice used for experiments (a key issue in BAT/browning biology). In the Methods, female are stated to be used for SVF preparations. Is this the same for the in vivo studies in the last part of the manuscript?

Reviewer #2:

Remarks to the Author:

Authors identify miR-327 as a regulator of beige adipogenesis. Using in vitro and in vivo experiments, they demonstrate that miR-327 negatively affects the browning, and claim that these effects are mediated by FGF10, a target of miR-327 identified in the study. The work is novel in a sense that it suggests the browning being regulated by autocrine signalling mediated by a growth factor under the regulation of a miRNA, it self physiologically regulated (by cold). The main weakness of the study is that the causative link between miR-327 and FGF10 in regulating the browning is not confirmed. Addressing the several additional points indicated bellow is needed to strengthen the story and provide the physiological importance of the discoveries.

1. The role of FGF10 in regulating fat morphology has been suggested previously. From the present study, it is unclear whether the Fgf10 preferentially favours differentiation or increases proliferation. Different confluences affect the differentiation largely, and these two potential effects should be discriminated.
2. Similarly, the proliferation of the preadipocytes during miR-327 overexpression or inhibition is not addressed. The authors should quantify the number of cells and proliferative markers at several time points during the differentiation assays. In addition, UCP1 and the rest of the browning markers should be shown in 1-2 more time points during differentiation.
3. The key evidence of the direct link between FGF10 and miR-327 comes from the supplementation of FGF10 recombinant protein after miR-327 overexpression in vitro. Authors should comment whether this is done in physiological doses. Either way, enhanced browning after FGF10 supplementation is also seen in the control cells which already have some FGF10 present, thus making the interpretation of this experiment difficult. To strengthen this potential link, I find it necessary that miR-327 and FGF10 are co-inhibited and the differentiation rates measured. In addition, authors should examine if conditioned medium after miR327 inhibition is sufficient to promote browning (in primary and/or 3T3-L1 cells). As the FGF10-KO mice exist, authors could test the miR327 – FGF10 link using this animal model.
4. Further to point 3, would blocking FGFR2 abolish the browning induced by miR-327 inhibition?
5. The overall phenotype of the mice after miR-327 overexpression or inhibition needs to be presented – including total body weight, food intake, glucose tolerance, insulin sensitivity, cold tolerance etc. Authors should also show the fat mass (total and of injected fat pads), and determine if the total number of adipocytes is changed.

Minor Points:

1. The authors interpret the data assuming that dominant or only path for gaining beige adipocyte is a pre-adipocyte differentiation. The authors should comment on how does their study relate to the mature adipocyte interconversion recently suggested (Rosenwald et al., 2013, NCB).
2. Page 13, Line 19 (“we measured changes OF THE energy...”)
3. Page 3, Line 12 - references should be added.

Reviewers' comments:

Reviewer #1 (Remarks to the Author):

General comment:

The manuscript by Fischer et al. reports the identification of miR-327 as a regulatory factor in the control of browning of adipose tissues. It claims that the effects of miR-327 (a negative regulator) occur via the induction of FGF10 which acts in an autocrine manner. The study provides interesting data and add one more actor to the growing list of small regulatory RNAs involved in the control of brown/beige adipose biology.

Weaknesses of the manuscript are: the lack of identification of the cell types accounting for some of the phenomena observed in the rather heterogeneous SVF fraction of adipose tissue, lack of measurement of the extracellular release of FGF10, necessity of a stronger demonstration that it is effectively FGF10 the major mediator of miR-327 effects, and necessity of a more thorough assessment of the systemic alterations in the in vivo models reported in the last part of the manuscript.

Response: We thank the reviewer for his/her precise understanding of our findings. We agree with the reviewer's comments, which are very valuable. According to the reviewer's comments, we have performed new experiments by fractionating different cell types using specific markers. To gain detailed information about the cellular source of FGF10, we isolated different cell types from the SVF of scWAT. These cell types include CD45⁺ immune cells (T cells, monocytes, leukocytes, dendritic cell and NK-cells), CD31⁺ vascular endothelial cells, PDGFR α ⁺ stromal fibroblasts, mature adipocytes, and the rest of PDGFR α ⁻ cells. Interestingly, Fgf10 mRNA expression was predominant in the PDGFR α ⁺ cellular fraction (Fig. 3g). In contrast, CD45⁺, CD31⁺, adipocyte and PDGFR⁻ fractions expressed negligible levels of Fgf10 mRNA. Also, the expression level of Fgfr2 mRNA in the PDGFR α ⁺ fraction was 4-20-fold higher than in other fractions (Fig. 3h). These findings suggest the existence of an autocrine regulatory loop in the PDGFR α ⁺ positive cells. Consistently, miR-327 was also found highest in PDGFR α ⁺ positive cells (Fig. 3i), supporting the regulation of its target gene in preadipocytes. These new data were included in Fig. 3g-i of the revised manuscript.

According to the reviewer's comment, we have also measured the extracellular levels of FGF10 protein released from primary preadipocytes. We have performed a quantitative ELISA assay to detected intracellular and extracellular FGF10 protein levels in primary preadipocytes. Primary preadipocytes were isolated from the subcutaneous WAT and cultured for 72 h and conditioned medium were collected. As shown in Fig. 3j, a nearly 7-fold higher FGF10 protein level was detected in the conditioned medium relative to its intracellular amount. These findings demonstrate that FGF10 is effectively exported from its producing cells. These new data were included in Fig. 3j of the revised manuscript.

We have also experimentally addressed the last two points from the reviewer (see below).

Specific points:

Comment: • Why data obtained in Fig 2 prioritized the use of visceral WAT, where the browning process is much weaker than scWAT?

Response: The reason why we chose to use visceral WAT in Fig. 2 was because

subcutaneous WAT under room temperature is already modestly browning which generates higher variations and background between experimental animals. Because visceral WAT shows less variation between each individual mouse and demonstrates a strong browning phenotype when treated with CL-316243, more stable experimental data could be obtained. Thus, we chose visceral fat for this experiment to get some clues of browning-related microRNAs and their targeted genes. Of course, all these initial findings were validated in subcutaneous WAT as demonstrated in our manuscript. We thank the reviewer for his/her understanding.

Comment: • The Results section in page 7 refers systematically to data shown in Fig2 as tissues (scWAT, vsWAT, iBAT) whereas data correspond to SVF isolated from tissues. The paragraphs should be written in a way that it states more explicitly that they correspond to SVF from tissues. By the way, providing data of the actors of the system proposed (miR-327, FGF10, FGFR2) in the tissue/mature fraction would be of interest to assess the specificity of the effects found in non-adipose versus mature adipose fractions, even if basal levels could be lower.

Response: Completely agree. We have rephrased the paragraph in Page 7 in such a way to explicitly indicate the SVFs from their corresponding tissues. According to the reviewer's suggestion, we have performed new experiments to quantitatively measure miR-327, FGF10, and FGFR2 in the mature adipocyte fractions of scWAT and compared them with the levels measured in other cellular fractions in this tissue. Expectedly, these actors were expressed at very low levels in the PDGFRA- fractions (see Fig. 3g-i of the revised manuscript). We further isolated mRNAs from mature adipocytes derived from scWAT, visWAT and iBAT under thermoneutral 30 °C- and cold-exposed conditions. As shown in Supplementary Fig. 3a-d, expression levels of miR-327, Fgf10 and Fgfr2 remained unchanged in the mature adipose fractions under 4 °C exposure. Thus, the miR-327-FGF10-FGFR2 regulatory pathway is restricted and specific for the non-adipose fractions. We have included these new data in Supplementary Fig. 3a-d of the revised manuscript.

Comment: • Again, for consistency in the presentation of data: Fig2 should indicate explicitly when data correspond to mRNA (panel j to m). Otherwise, in Fig 3, panels a and b appear to correspond to data on whole tissue, not SVF. What's the reason for not determining the FGF10 protein in SVF as in Fig 2? The interpretation of data may be rather affected. What about the influence/ behavior of the mature cell fraction, excluded in the mRNA measurements in Fig 2 but present in the protein measurements in Fig 3?

Response: Agree. The reviewer is completely correct. We apologize for not clearly indicate the mRNA of SVFs isolated from various adipose depots. In the revised manuscript, we have labeled mRNA in panels j-m of Fig. 2. It is correct that FGF10 protein levels presented in Fig. 3a-c of the revised manuscript were obtained from the whole tissues but not SVFs. As shown in Fig. 3j of the revised manuscript, the majority of FGF10 protein molecules is released from its biosynthetic cells, therefore measurement of FGF10 protein in SVF would not reflect the total levels. It is known that FGF10 displays a high affinity for heparin (Patel et al., J Biol Chem. 2008 Apr 4; 283(14): 9308–9317). Upon secretion, FGF10 protein would bind to the heparan sulfate proteoglycans in the extracellular matrix. Thus, we believe that it is more accurate to measure FGF10 protein levels using the whole tissue even though FGF10 is produced from cells in the SVF.

In this regard (see above), it is not surprising that FGF10 mRNA is rather limited to SVFs

whereas its protein can be spread to other cell surface or extracellular matrix upon its release. Nevertheless, FGFR2 is also relatively specifically expressed in SVFs (more explicitly in the PDGFR α ⁺ fraction) (see Fig. 3h of the revised manuscript). Thus, the interpretation of autocrine stimulation is supportively evident.

Comment: • Claims for paracrine effects of FGF10 should include measurements of the FGF10 protein in the extracellular compartment in several experimental settings. The absence of these data is a weakness of the manuscript as it claims explicitly for autocrine actions.

Response: Once FGF10 is released from its synthetic cells (more restricted in the stromal PDGFR α ⁺ preadipocytes), it can bind to the extracellular matrix and cell surface of other cells. In this regard, it is possible to undergo a paracrine regulatory mechanism. However, the FGF10 receptor FGFR2 has to be expressed in that cell type in order to respond to FGF10. Although we cannot completely exclude the existence of paracrine mechanisms, it is unlikely that a paracrine regulation would play a major role. Our supportive evidence include: 1) FGF10 is mainly produced from PDGFR α ⁺ preadipocytes; 2) FGFR2 is also mainly expressed in PDGFR α ⁺ preadipocytes; 3) miR-327 is mainly expressed in PDGFR α ⁺ preadipocytes; and 4) Differentiation of pure 3T3L1 cell population into adipocytes can be blocked by siFGF10 (see below).

Nevertheless, we analyzed FGF10 protein levels in the extracellular and intracellular compartments of preadipocytes and mature adipocytes in several experimental settings. Again, FGF10 protein is mainly localized in the extracellular compartment of pre-adipocytes. These data support our conclusions. We have included these new data in Supplementary Fig. 3j and k of the revised manuscript. To tone down our conclusions, we have discussed the possibility of existence of a paracrine mechanism in the discussion section.

Comment: • The experiment shown in Fig 4h is essential to demonstrate that the effect of miR-327 down-regulation leading to promotion of adipogenic differentiation and the effect of FGF10 promoting also differentiation were two linked phenomena, and not two independent processes occurring in parallel. The statements that FGF10 rescues miR-327 driven impairment in differentiation should be in some way toned down according to Fig 4h. In fact, the fold-induction in differentiation elicited by FGF10 in the absence of miR-327 (the two NC panels) appears not so different from that elicited in the presence of miR-327 (the two miR-327 panels). The conclusive sentence in this regard in lines 244, 245 and the heading in 224 should be toned down accordingly. Moreover, models using loss-of-function for FGF10 would be needed to conclusively rule out that miR-327 is not acting through pathways other than FGF10.

Response: Agree. We have changed the heading of 224 and deleted the conclusive sentence in lines of 244 and 245. We thank the reviewer for pointing this out.

According to the reviewer's recommendation, we have performed new experiments using a combination approach by knocking down FGF10 in the presence and absence of miR-327 inhibitor. As shown in Fig. 5b of the revised manuscript, introduction of a miR-327 inhibitor to 3T3-preadipocytes permitted differentiation into mature adipocytes. However, knockdown of Fgf10 by a siFgf10 virtually completely blocked the miR-327-inhibitor-induced adipocyte differentiation. The blocking effect was rescued by FGF10-containing conditioned medium derived from untreated preadipocytes (Fig. 6g and h of the revised manuscript). These findings demonstrate that miR-327 modulates differentiation of preadipocytes into adipocytes

by altering FGF10 levels.

Comment: • Also in relation to the experiments depicted in Fig 4, in fact the data provided in a-j panels concern general adipogenesis (3T3-L1 are not cell models of beigeing). Does it mean that miR-327 affects adipogenic differentiation regardless of whether this is “white” differentiation or “beige” differentiation? If cell models and *in vivo* models were used to check for white adipogenic differentiation instead of the models of browning used across the paper and they were challenged for miR-327 manipulation, would conclusions be the same?

Response: We agree with the reviewer’s view. 3T3-L1 cells are not explicitly committed for browning adipocyte differentiation. We believe that driving differentiation from preadipocytes to mature adipocytes is a common process regardless if it is “white” or “beige” differentiation. However, under browning conditions such as cold exposure or β 3-adrenergic activation, the differentiation is drifted towards a beige direction. miR-327 plays an essential role for controlling the FGF10-induced preadipocyte differentiation. In our *in vivo* models, the experimental setting favors browning adipocyte differentiation and miR-327 blocks this process and thus impairs the browning phenotype. Thus, our conclusions remain the same.

Comment: • Proper assessment of the data obtained in the experiment shown in panels m and n of Fig 4 (transcript levels in the presence of isoproterenol) should include the basals, showing data only “in the presence of Isoproterenol”, without the basals of untreated cells, makes no possible a precise interpretation of the data.

Response: Completely agree. We thank the reviewer for this important comment and we apologize for not including the basals in the original version of the manuscript. We have now included the basal levels of the original Fig. 4m and n. These revised figures are now presented as Fig. 5g and h of the revised manuscript.

Comment: • Why no data are provided on the FGF10/FGFR2 system in the *in vivo* experiments shown in Fig 5? This is needed for consistency in the core scientific message of the manuscript.

Response: According to the reviewer’s suggestion, we have performed new experiments to measure FGF10 in the *in vivo* experimental settings. Expectedly, adenoviral transduction of miR-327 effectively blocked FGF10 expression in cold- and CL-treated browning scWAT. Similarly, Ad-miR-327 markedly decreased UCP1 expression, validating the significant impairment of the browning phenotype. We have included these new data in Fig. 8a-g of the revised manuscript.

Comment: • In relation to the experiments depicted in the last part of Fig 6., considering that the administration of the miR-327 inhibitor is not adipose-driven, systemic data in these mice in addition to oxygen consumption should be provided to properly assess the direct versus indirect effects leading to the differences in WAT browning observed. Moreover, iBAT behavior (as responsible for most adaptive thermogenesis activity in mice) in response to the inhibitor should be shown. Finally, the changes of NE-induced oxygen consumption in mice should be shown in more detail, the data on NE-induced oxygen consumption should be accompanied by the oxygen consumption data in basal, before NE-treatment, in the miR-327 gain- and loss-of-function manipulations in mice. The significant but rather mild effects found between groups require a precise assessment of these data.

Response: Agree. We have now included the systemic data related to original Fig. 5 and Fig. 6 (now Fig. 7 and Fig. 9) in Supplementary Fig. 6 and Supplementary Fig. 8 of the revised manuscript. We have also included the iBAT data in Supplementary Fig. 7 of the revised manuscript. Again, in these *in vivo* experimental settings, we did not see any changes in iBAT morphology or activation. According to the reviewer's recommendation, we included oxygen consumption in miR-327 mimic and inhibitor treated mice during daytime and nighttime in addition to NE-induced oxygen consumption (See Fig. 8j-l and Fig. 9j-l of the revised manuscript).

Other specific points are:

Comment: Some of the statements in Introduction about the physiological role of WAT browning (lines 68-72) should be toned down, the relative physiological role of WAT browning is rather controversial (see Nedergaard et al. Cell Metab. 2014 20:396-407, or Nedergaard et al. Biochim Biophys Acta. 2013 1831:943-9).

Response: We agree. In the revised manuscript, we have toned down the sentence (lines 68-72 of the original manuscript) and have cited Nedergaard et al. Cell Metab. 2014 20:396-407.

Comment: To be strict, mention on how miRNAs control protein production should include inhibition of translation, in addition to mRNA down-regulation (lines 85-87)

Response: Completely agree. In the revised manuscript, we have included inhibition of translation to precisely describe miRNA's functions.

Comment: The use of a highly artificial system for the assessment of the targeting of miRNA327 upon FGF10 mRNA (including the use of HEK297 cells, of renal origin) should lead to tone down the conclusion in this regard, as the whole rationale of the manuscript concerns the pre-adipocyte/adipocyte context.

Response: Completely agree. Since this cell line was only used for target validation and the 3'UTR of Fgf10 was cloned into a dual luciferase promoter, changes in luciferase activity were the result of miR-327 mimics binding to the 3'UTR of Fgf10 on the transiently transfected promoter. Therefore, any cell line that takes up the plasmid and miR-327 mimics in their cytoplasm should show similar results. In other words, these cells were used as molecular tools, but not to define any of the adipose-related functional outcomes. Nevertheless, we have toned down our finding by deleting the conclusive sentence.

Comment: Fig 2. Differences in gene expression by comparing SVFs can be affected by differential cell composition (biological) but also to technical aspects of preparation by using as sources for isolation tissues with rather different in structure (because of being control tissues or CL-treated tissues). Please comment.

Response: We thank the reviewer for his/her tremendous experiences in this field. We agree. Most cell types of the SVF of adipose tissue undergo changes during browning. These changes will certainly influence the gene expression profile. To the best of our knowledge, it is impossible to stably control these changes in any experimental settings. One of the important approaches to resolve this issue is to sort out different cell types, which were subjected for further analysis as shown in Fig. 3g-i of the revised manuscript.

Comment: Representation of phosphorylation of Akt and ERK is unclear. The ratio between phosphorylated and total protein should be shown. Otherwise, Page 10 states no significant differences in P-ERK, but a $P < 0.001$ difference (minor, but apparently highly significant) is shown in Fig3 q (?). Providing the phosphorylated/non-phosphorylated ratios may help for clarity.

Response: We apologize for this mistake. There is no significant difference in P-ERK or ERK protein levels after miR-327 mimic treatment. P value of < 0.001 related stars were apparently mislabeled in original Fig. 3r. We have now corrected this error. These data are now presented in Fig. 4k and l of the revised manuscript.

Comment: The heading of the last Fig. should be changed, as it includes not only data with the miR-327 inhibitor but also with the adenoviruses used in the previous Figure. Perhaps the oxygen consumption data in the adenovirus-treated mouse model should be moved to the previous Figure.

Response: We thank the reviewer for this excellent suggestion. The last figure is now reorganized and heading reflects the figure composition. Oxygen consumption data for the adenovirus-treated mouse model are now shown in Fig. 8j-l of the revised manuscript.

Comment: Write in full the number after CL in the Figure legends

Response: Thank you. Corrected.

Comment: In Fig2, indicate explicitly when data correspond to mRNA (panel j to m).

Response: Agree. We have revised Fig. 2 according to the reviewer's suggestion.

Comment: Please, provide details on sex of mice used for experiments (a key issue in BAT/browning biology). In the Methods, female are stated to be used for SVF preparations. Is this the same for the in vivo studies in the last part of the manuscript?

Response: We thank the reviewer for this important suggestion. We have used female mice throughout the entire experiments. We have indicated this in the Methods section.

Reviewer #2 (Remarks to the Author):

Comment: Authors identify miR-327 as a regulator of beige adipogenesis. Using in vitro and in vivo experiments, they demonstrate that miR-327 negatively affects the browning, and claim that these effects are mediated by FGF10, a target of miR-327 identified in the study. The work is novel in a sense that it suggests the browning being regulated by autocrine signalling mediated by a growth factor under the regulation of a miRNA, it self physiologically regulated (by cold). The main weakness of the study is that the causative link between miR-327 and FGF10 in regulating the browning is not confirmed. Addressing the several additional points indicated bellow is needed to strengthen the story and provide the physiological importance of the discoveries.

Response: We thank the reviewer for considering novelty of our work, especially the autocrine signaling. We certainly agree with the reviewer for further mechanistic

characterization of the causative link between miR-327 and FGF10.

Major points

Comment: 1. The role of FGF10 in regulating fat morphology has been suggested previously. From the present study, it is unclear whether the Fgf10 preferentially favors differentiation or increases proliferation. Different confluences affect the differentiation largely, and these two potential effects should be discriminated.

Response: We thank the reviewer for raising this very interesting question. According to the reviewer's suggestion, we have performed new experiments by treating preadipocytes with FGF10, siFgf10, miR-327, and miR-327 inhibitor and measuring cell viability and proliferation before, during and after differentiation. The proliferation assay was used to determine relative viability in cells after different treatments. The absorbance at 490 nm reflects the number of living cells per well. No significant difference in viable cells could be measured within the groups before and during differentiation. These findings demonstrate that cell proliferation and cell number are unaffected by different treatments (see Supplementary Fig. 4 a-c of the revised manuscript). This notion was further validated by Ki67 staining (Supplementary Fig. 4d-g of the revised manuscript). Thus, FGF10 is critically involved in preadipocyte differentiation rather than proliferation and the confluency before and during differentiation is similar in all experimental settings.

Comment: 2. Similarly, the proliferation of the preadipocytes during miR-327 overexpression or inhibition is not addressed. The authors should quantify the number of cells and proliferative markers at several time points during the differentiation assays. In addition, UCP1 and the rest of the browning markers should be shown in 1-2 more time points during differentiation.

Response: Agree. We have performed cell proliferation assay in the presence and absence of miR-327 mimic and inhibitor treatment. In undifferentiated preadipocytes, miR-327 overexpression and inhibition had no impacts on cell proliferation (Supplementary Fig. 4a, d and e of the revised manuscript). After two days differentiation, preadipocyte proliferation remained unaffected by miR-327 overexpression and inhibition (Supplementary Fig. 4b, d and f of the revised manuscript). At day 6, preadipocytes underwent terminal differentiation without further proliferation. The number of proliferating cells in siFgf10- and miR-327-treated cells were higher than their respective controls (Supplementary Fig. 4c, d, g). SiFgf10- and miR-327 blocked cell differentiation to become mature adipocytes, thus continuing to allow cell proliferation. However, these proliferative effects were not produced by FGF10 per se because FGF10 does not affect cell proliferation in undifferentiated cells. Knocking down FGF10 in preadipocytes would block their differentiation and would therefore allow cells to continue proliferate. Thus, proliferation and differentiation is a balanced scenario, the former, however, doesn't seem to be directly affected by FGF10 or miR-327 levels.

According to the reviewer's suggestion, we have analyzed UCP1 and other browning markers in 3T3-L1 cells before, during and after differentiation. As shown in Supplementary Fig. 3f-i, UCP1 and other browning markers were gradually increased during differentiation of isoproterenol treated cells. miR-327 largely ablated expression levels of UCP1 and other browning markers in differentiated cells. These new findings are now included in Supplementary Fig. 3f-i of the revised manuscript.

Comment: 3. The key evidence of the direct link between FGF10 and miR-327 comes from the supplementation of FGF10 recombinant protein after miR-327 overexpression in vitro. Authors should comment whether this is done in physiological doses. Either way, enhanced browning after FGF10 supplementation is also seen in the control cells which already have some FGF10 present, thus making the interpretation of this experiment difficult. To strengthen this potential link, I find it necessary that miR-327 and FGF10 are co-inhibited and the differentiation rates measured. In addition, authors should examine if conditioned medium after mir327 inhibition is sufficient to promote browning (in primary and/or 3T3-L1 cells). As the FGF10-KO mice exist, authors could test the miR327 – FGF10 link using this animal model.

Response: Agree. According to the reviewer's suggestion, we performed new experiments to simultaneously inhibit miR-327 and FGF10 and measured differentiation rates. As shown in Fig. 5b of the revised manuscript, introduction of a miR-327 inhibitor to 3T3-preadipocytes permitted differentiation into mature adipocytes. However, knockdown of Fgf10 by a siFgf10 virtually completely blocked the miR-327-inhibitor-induced adipocyte differentiation as shown in Fig. 6g and h of the revised manuscript. Thus, FGF10 seems to be the dominant factor that drives miR-327 inhibitor treated 3T3-L1 cells into differentiation. These findings demonstrate that miR-327 acts through an FGF10-triggered pathway.

According to the reviewer's suggestion, we have performed new experiments using conditioned medium from miR-327-inhibitor-treated cells. As shown in Fig. 6g and h of the revised manuscript, conditioned medium from miR-327-inhibitor-treated cells could rescue preadipocyte differentiation in siFgf10-treated cells.

Indeed, we have quantitatively measured FGF10 protein in the conditioned medium of miR-327-inhibitor-treated cells using a sensitive ELISA assay as shown in Supplementary Fig. 3j. The FGF10 levels in miR-327-inhibitor-treated preadipocytes were significantly higher relative to its intracellular counterpart. After 72 h incubation, preadipocytes produced between 2-9 ng ml⁻¹ of extracellular FGF10 protein in the conditioned medium under different experimental treatments, which is in a similar range as the extraneous added recombinant protein (10 ng ml⁻¹). We believe that after prolonged incubation, these preadipocytes would produce even higher amount of FGF10. Therefore, we believe that the amount of FGF10 added to the cell culture is physiologically relevant.

Regarding FGF10-KO mice, Fgf10 is essential for limb and lung formation (Nature Genetics 21, 138 - 141 (1999) doi:10.1038/5096). Fgf10^{-/-} mice die at birth due to the lack of lung development. This early time point of embryonic death makes it impossible to study browning of adipose tissue (white adipose tissue develops mainly after birth).

To experimentally address this important issue, we used Fgf10 heterozygous mice for further experiments. We have found that FGF10 protein levels were very low in Fgf10 heterozygous mice relative to those in wildtype mice (Supplementary Fig. 9a of the revised manuscript). Treatment of Fgf10^{+/-} mice with the miR-327 inhibitor did not significantly augment a browning phenotype. miR-327 inhibitor-treated scWAT lacked UCP1 and COX4 expression. Similarly, no significant difference in energy expenditure was recorded in Inh-miR-327-treated Fgf10^{+/-} mice compared to Inh-miR-NC Fgf10^{+/-} mice. Both, FGF10 and UCP1 protein levels were very low and barely increased after inhibitor treatment. Thus, this genetic approach further strengthens our conclusions that miR-327 regulates FGF10 levels and thereby regulates WAT browning. We have included these new data in Fig. 10 and

Supplementary Fig. 9 of the revised manuscript.

Comment: 4. Further to point 3, would blocking FGFR2 abolish the browning induced by miR-327 inhibition?

Response: We thank the reviewer for this excellent suggestion. We have knocked down *Fgfr2* and simultaneously treated with the miR-327 inhibitor. *Fgfr2* inhibition effectively blocked the miR-327 inhibition-induced preadipocyte differentiation. We have included these new data in Fig. 6i and j of the revised manuscript.

Comment: 5. The overall phenotype of the mice after miR-327 overexpression or inhibition needs to be presented – including total body weight, food intake, glucose tolerance, insulin sensitivity, cold tolerance etc. Authors should also show the fat mass (total and of injected fat pads), and determine if the total number of adipocytes is changed.

Response: Agree. We have now included these overall phenotypes including total body weight, food intake, scWAT weight, visWAT weight, iBAT weight, glucose tolerance, HOMA-IR, basal metabolic rates, NE-induced respiration and core body temperature for all *in vivo* experiments in Supplementary Fig. 6, 8 and 9 and in the main Fig. 8h-l, Fig. 9i-l and Fig. 10 i, j).

Minor points

Comment: 1. The authors interpret the data assuming that dominant or only path for gaining beige adipocyte is a pre-adipocyte differentiation. The authors should comment on how does their study relate to the mature adipocyte interconversion recently suggested (Rosenwald et al., 2013, NCB).

Response: Agree. In our experimental settings, we show that differentiation of preadipocytes to beige adipocytes is a key mechanism. However, we cannot exclude other experimental settings in which interconversion between mature adipocytes plays a role as an alternative mechanism. We have discussed this possibility in the revised manuscript.

Comment: 2. Page 13, Line 19 (“we measured changes OF THE energy...”)

Response: Thank you. We have corrected this error in the revised manuscript.

Comment: 3. Page 3, Line 12 - references should be added.

Response: Agree. We have included the reference by Cohen et al., *Cell*, 156, 304-316, 2014 in page 3 of the revised manuscript.

Reviewers' Comments:

Reviewer #1:

Remarks to the Author:

The revised manuscript by Fischer et al. contains a number of additional experiments that complement the initial findings and fulfill most of my former criticisms. New data appear convincing and add to the manuscript a remarkable strength. Additions and modifications to the manuscript also provide a more balanced interpretation of data.

My only minor comment is to suggest an addition to the Discussion stating that some of the mechanisms depicted here might be relevant not only to the browning process but also to overall adipogenic differentiation, as evidenced by some of the effects found systematically in the 3T3L1 model.

Reviewer #2:

Remarks to the Author:

Authors have addressed my comments.

Response to Reviewer's Comments

Thank you for your decision letter on our manuscript (NCOMMS-17-05062-T) entitled "A miR-327-FGF10–FGFR2-mediated autocrine mechanism controls white fat browning". We are extremely happy to see your positive decision. Thank you so much for handling our manuscript.

Our point-by-point responses to the editor and reviewers' comments are as follows:

Reviewer #1 (Remarks to the Author):

Comment: The revised manuscript by Fischer et al. contains a number of additional experiments that complement the initial findings and fulfill most of my former criticisms. New data appear convincing and add to the manuscript a remarkable strength. Additions and modifications to the manuscript also provide a more balanced interpretation of data. My only minor comment is to suggest an addition to the Discussion stating that some of the mechanisms depicted here might be relevant not only to the browning process but also to overall adipogenic differentiation, as evidenced by some of the effects found systematically in the 3T3L1 model.

Response: We thank the reviewer for this excellent suggestion. In the revised manuscript, we have extended our discussion by stating that this mechanism also applies to other overall adipogenic differentiation.